# Automated Filtering of Human Feedback Data for Aligning Text-to-Image Diffusion Models

**Yongjin Yang**[1*]  **Sihyeon Kim**[1*]  **Hojung Jung**[1]  **Sangmin Bae**[1]

**SangMook Kim**[2]  **Se-Young Yun**[1†]  **Kimin Lee**[1†]

KAIST AI[1]    Department of AI, Chungnam National University[2]

{dyyjkd, sihk, yunseyoung, kiminlee}@kaist.ac.kr

## Abstract

Fine-tuning text-to-image diffusion models with human feedback is an effective method for aligning model behavior with human intentions. However, this alignment process often suffers from slow convergence due to the large size and noise present in human feedback datasets. In this work, we propose **FiFA**, a novel automated data filtering algorithm designed to enhance the fine-tuning of diffusion models using human feedback datasets with direct preference optimization (DPO). Specifically, our approach selects data by solving an optimization problem to maximize three components: preference margin, text quality, and text diversity. The concept of preference margin is used to identify samples that are highly informative in addressing the noisy nature of feedback dataset, which is calculated using a proxy reward model. Additionally, we incorporate text quality, assessed by large language models to prevent harmful contents, and consider text diversity through a k-nearest neighbor entropy estimator to improve generalization. Finally, we integrate all these components into an optimization process, with approximating the solution by assigning importance score to each data pair and selecting the most important ones. As a result, our method efficiently filters data automatically, without the need for manual intervention, and can be applied to any large-scale dataset. Experimental results show that **FiFA** significantly enhances training stability and achieves better performance, being preferred by humans $17\%$ more, while using less than $0.5\%$ of the full data and thus $1\%$ of the GPU hours compared to utilizing full human feedback datasets. Warning: This paper contains offensive contents that may be upsetting.

## 1 Introduction

Large-scale models trained on extensive web-scale datasets using diffusion techniques (Ho et al., 2020; Song et al., 2020), such as Stable Diffusion (Rombach et al., 2022), Dall-E (Ramesh et al., 2022), and Imagen (Saharia et al., 2022), have enabled the generation of high-fidelity images from diverse text prompts. However, several failure cases remain, such as difficulties in illustrating text content or incorrect counting (Lee et al., 2023). Fine-tuning text-to-image diffusion models using human feedback has recently emerged as a powerful approach to address this issue (Black et al., 2023; Fan et al., 2024; Prabhudesai et al., 2023; Clark et al., 2023). Unlike the conventional optimization strategy of likelihood maximization, this framework first trains reward models using human feedback (Kirstain et al., 2024; Wu et al., 2023; Xu et al., 2024) and then fine-tunes the diffusion models to maximize reward scores through policy gradient (Fan et al., 2024; Black et al., 2023) or reward-gradient based techniques (Prabhudesai et al., 2023; Clark et al., 2023). More recently, Diffusion-DPO (Wallace et al., 2023), which directly aligns the model using human feedback without the need for training reward models, has been proposed. This approach enables fine-tuning diffusion models at scale using human feedback, with the additional benefit of leveraging offline datasets more effectively.

However, fine-tuning diffusion models using human feedback requires considerable time and computational resources. For instance, even with the relatively efficient Diffusion-DPO (Wallace et al.,

---

*Equal contribution

†Corresponding authors

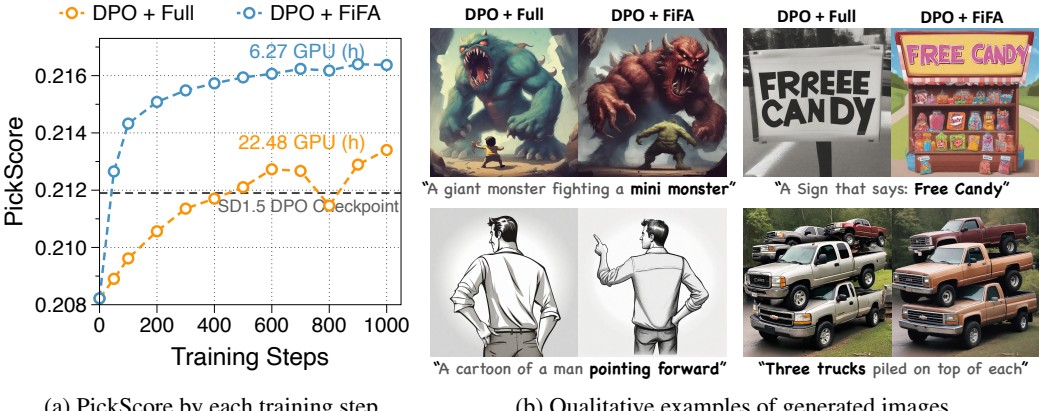

(a) PickScore by each training step      (b) Qualitative examples of generated images

Figure 1: (a) PickScore (Kirstain et al., 2024) at each training step of the SD1.5 model using data filtered with **FiFA**, which uses $0.5\%$ of the data, compared to the model trained with full dataset. Our method significantly outperforms the alternative, converging faster while requiring about 4x fewer GPU hours to match the performance of the SD1.5-DPO released checkpoint[1]. (b) Qualitative evaluation of training on the full data and data selected with our **FiFA** for various prompts.

2023), it still takes more than thousands of GPU hours to fully fine-tune SDXL model Podell et al. (2023) on the large-scale Pick-a-Pic v2 dataset (Kirstain et al., 2024). This is attributed to the multiple denoising processes involved in diffusion models, as large diffusion models must be trained on multiple timesteps (Fan et al., 2024; Prabhudesai et al., 2023; Clark et al., 2023). Moreover, the noisy nature of feedback dataset slows down the convergence speed by making it harder for the model to accurately fit user preferences (Yang et al., 2023b; Chowdhury et al., 2024).

Some model-centric approaches such as model pruning (Fang et al., 2023; Ganjdanesh et al., 2024), which reduces the size of diffusion models, or alternative scheduling techniques (Luo et al., 2023), which aim to reduce the number of timesteps, are utilized to improve efficiency. However, the nature of large-scale feedback datasets diminishes the impact of these methods, as the training cost increases dramatically with the growth of both model size and dataset, despite improvements in efficiency per data point. Dai et al. (2023) have attempted to create smaller datasets manually to reduce the need for large-scale data, but this approach requires significant human effort and is not applicable for reducing the size of large-scale feedback datasets, which may serve different purposes. This highlights the need for an automated approach to extract meaningful subsets from large-scale feedback data.

In this paper, we propose a novel automated **Fi**ltering framework that selectively integrates human **F**eedback, designed for efficient **A**lignment of diffusion models (**FiFA**). We frame the filtering task as an optimization problem, aiming to find a subset that maximizes the three components; (1) preference margin, (2) text quality, and (3) text diversity. A key component of our optimization is selecting data pairs that are more informative, as determined by their preference margins, which are calculated using a proxy reward model. Specifically, training pairs with a low preference margin can be considered noisy and ambiguous data, as their preferences may easily flip, thereby hindering the training process (Chowdhury et al., 2024; Yang et al., 2023b; Rosset et al., 2024). Furthermore, to address the concerns on harmfulness problems and coverage of selected subset induced by relying only on preference margin, we also consider the quality and diversity of the text prompts in the objective function. We assess text quality using a Large Language Model (LLM), following Sachdeva et al. (2024), and measure text diversity by calculating the entropy of embedded text prompts (Zheng et al., 2020) approximated using a k-nearest neighbor estimator (Singh et al., 2003). To integrate all these components, we define an objective function that combines the three metrics into a single optimization problem. Additionally, to improve efficiency, we approximate the solution by assigning a data importance score for each data pair, making **FiFA** efficient and applicable to large-scale datasets through an automated process.

---

[1] https://huggingface.co/mhdang/dpo-sd1.5-text2image-v1

In our experiments with open-sourced text-to-image diffusion models Stable Diffusion 1.5 (SD1.5) and Stable Diffusion XL (SDXL) (Podell et al., 2023), **FiFA** significantly improves training efficiency compared to fine-tuning with full datasets. As shown in Figure 1a, by using only $0.5\%$ of the full Pick-a-Pic v2 dataset (Kirstain et al., 2024), the SD1.5 model trained using **FiFA** demonstrates a significantly faster increase in PickScore (Kirstain et al., 2024) than the SD1.5 model trained on the full dataset. Moreover, the SDXL model trained using **FiFA** is preferred $17\%$ more than the model trained with the full dataset by human annotators when evaluated on the HPSv2 benchmark (Wu et al., 2023), with the preferred images showing better text-image alignment and higher quality, as illustrated in Figure 1b. We remark that this is achieved while requiring less than $1\%$ of the GPU hours. Additionally, **FiFA** reduces harmfulness by more than $50\%$ for neutral prompts compared to using full dataset, by prioritizing text quality. Overall, our proposed **FiFA** allows large text-to-image diffusion models to be efficiently trained on a small yet important dataset, while showing better efficacy with high image quality.

## 2 PRELIMINARIES

**Diffusion Models**    Diffusion models (Ho et al., 2020) are probabilistic models that aim to learn a data distribution $p(\mathbf{x})$ by performing multiple denoising steps starting from a Gaussian noise. The diffusion process consists of two parts, forward process and backward process.

In the forward process, noise is progressively injected at each timestep $t$ according to $q(\mathbf{x}_t|\mathbf{x}_0) \sim \mathcal{N}(\sqrt{\bar{\alpha}_t}, (1 - \bar{\alpha}_t)\mathbf{I})$, where the noise schedule $\alpha_t$ is a monotonically decreasing function and $\bar{\alpha}_t := \prod_{s=0}^{t} \alpha_s$. The neural network $\boldsymbol{\epsilon_\theta}$ is trained to learn the denoising process with the following objective:

$$\mathcal{L}_{DM}(\theta) = \mathbb{E}_{\mathbf{x}_0, t}[\lambda(t)||\boldsymbol{\epsilon} - \boldsymbol{\epsilon_\theta}(\mathbf{x}, t)||_2^2], \tag{1}$$

where $\lambda(t)$ is determined by the noise schedule and $\boldsymbol{\epsilon}$ is a Gaussian noise. During generation, the diffusion model takes reverse denoising steps starting from a random Gaussian noise.

The conditional diffusion model (Rombach et al., 2022), such as a text-to-image diffusion model, aims to learn the data distribution $p(\mathbf{x}|\mathbf{c})$, trained using the conditional error $\boldsymbol{\epsilon}(\mathbf{x}, \mathbf{c})$ instead of the unconditional error $\boldsymbol{\epsilon}(\mathbf{x})$.

**Reward Learning in Text-to-Image Domains**    Using human preference data, the goal of reward learning is to train a proxy function aligned with human preferences. In text-to-image domains, given textual condition $\mathbf{c}$ and the generated image $\mathbf{x_0}$ from that condition, we assume a ranked pair with $\mathbf{x_0^w}$ as a "winning" sample and $\mathbf{x_0^l}$ with a "losing sample", that satisfy $\mathbf{x_0^w} > \mathbf{x_0^l}|\mathbf{c}$. Using the Bradley-Terry (BT) model, one can formulate maximum likelihood loss for binary classification to learn the reward model $r$, parameterized by $\phi$, as follows:

$$\mathcal{L}_{\text{BT}}(\phi) = -\mathbb{E}_{\mathbf{c}, \mathbf{x}_0^w, \mathbf{x}_0^l} \left[ \log \sigma(r_\phi(\mathbf{c}, \mathbf{x}_0^w) - r_\phi(\mathbf{c}, \mathbf{x}_0^l)) \right], \tag{2}$$

where $\sigma$ is a sigmoid function, $\mathbf{c}$ is a text prompt, and image pairs $\mathbf{x}_0^w$ and $\mathbf{x}_0^l$ labeled by humans.

**Direct Preference Optimization for Diffusion Models**    Direct Preference Optimization (DPO) (Rafailov et al., 2024) is an approach to align the model using human feedback without training a separate reward model. Directly applying DPO loss to diffusion models is not feasible, as an image is generated through the trajectory $\mathbf{x}_{T:0}$ where $T$ denotes the number of denoising steps, and obtaining the probability of this entire trajectory is generally intractable. Following Diffusion-DPO (Wallace et al., 2023), the DPO loss for diffusion models can be approximated as follows:

$$\mathcal{L}_{\text{DPO}}(\theta) = -\mathbb{E}_{t, \mathbf{c}, \mathbf{x}_0^w, \mathbf{x}_0^l} \log \sigma \left( -\beta T \omega(\lambda_t) \left[ \|\boldsymbol{\epsilon}^w - \boldsymbol{\epsilon}_\theta(\mathbf{x}_t^w, t)\|_2^2 - \|\boldsymbol{\epsilon}^w - \boldsymbol{\epsilon}_{\text{ref}}(\mathbf{x}_t^w, t)\|_2^2 \right. \right.$$
$$\left. \left. - \|\boldsymbol{\epsilon}^l - \boldsymbol{\epsilon}_\theta(\mathbf{x}_t^l, t)\|_2^2 + \|\boldsymbol{\epsilon}^l - \boldsymbol{\epsilon}_{\text{ref}}(\mathbf{x}_t^l, t)\|_2^2 \right] \right), \tag{3}$$

where $w(\lambda_t)$ is a weight function typically set to a constant, $\mathbf{x}_t^w, \mathbf{x}_t^l$ are the noised inputs of winning and losing images at timestep $t$ respectively, $\boldsymbol{\epsilon}^w, \boldsymbol{\epsilon}^l \sim \mathcal{N}(0, I)$ represent the Gaussian noise for the winning and losing images respectively, and $\boldsymbol{\epsilon}_{\text{ref}}$ is a pretrained diffusion model. Detailed derivation of DPO loss for diffusion is presented at Appendix C.

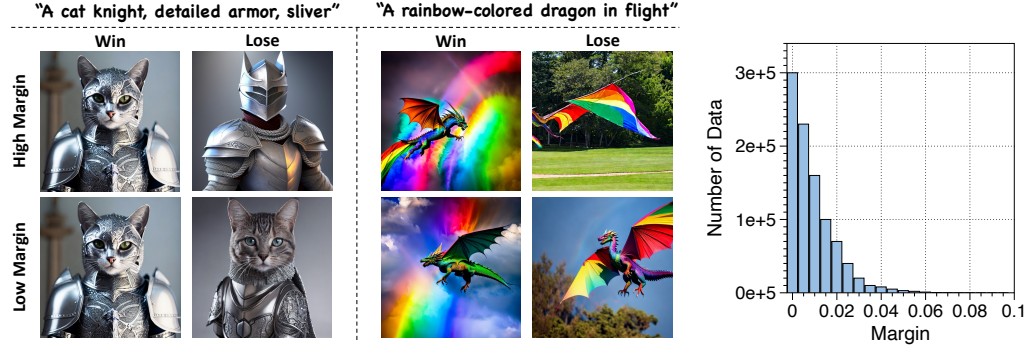

(a) Samples of high/low preference margin      (b) Distribution of reward margin

Figure 2: (a) Qualitative analysis of preference margin estimated through PickScore reward model. (b) Distribution of PickScore reward margins of Pick-a-Pic v2 train set.

# 3 METHODS

To address the inefficiency and noise in feedback datasets, we propose `FiFA`, which automatically filters the full human feedback data to obtain a subset for efficiently fine-tuning text-to-image models. Specifically, our method leverages preference margin as a key component to rapidly increase the reward value, while also considering the quality and diversity of the text prompts to mitigate harmfulness and ensure robustness. Additionally, we frame the task as a optimization problem to find the best subset that maximize these three components, resulting in an automated filtering framework applicable to any large-scale dataset.

## 3.1 PREFERENCE MARGIN

The noisy and ambiguous nature of human preference datasets has been well explored, where a labeled preference does not reflect the true preference and may contain spurious correlations (Yang et al., 2023b; Chowdhury et al., 2024). This can be especially problematic for the efficient fine-tuning of diffusion models, as such noisy data slow down training and reduce the generalization capability (Zhang et al., 2021). Inspired by recent papers that highlight the importance of clean preference pairs (Yang et al., 2023b; Rosset et al., 2024), we use the preference margin to enable more efficient and effective fine-tuning of diffusion models to alleviate this issue.

To estimate the preference margin between the winning and losing images, we utilize a proxy reward model $r_\phi$ trained on the full feedback dataset using the BT modeling approach with Eq. (2). This process does not pose efficiency concerns for text-image domains, as training a reward model using CLIP (Radford et al., 2021) or BLIP architectures (Li et al., 2022) demands significantly less time than training large diffusion models, and open-sourced text-image reward models like PickScore (Kirstain et al., 2024) and HPSv2 (Wu et al., 2023), trained on human feedback datasets, can also be utilized.

Figure 2a shows samples with different preference margins estimated using the PickScore proxy model. For example, given the prompt "A cat knight...", image pairs with high reward margin are more distinct, clearly showing that the image including a "cat" should be preferred over the one without it. On the other hand, for the low margin pairs, both images usually differ only in style, implying that preferences can be flipped depending on the annotator. This demonstrates that selecting pairs with clear distinctions offer more informative preferences based on the prompt.

Additionally, Figure 2b demonstrates that most pairs in the large-scale Pick-a-Pic v2 dataset are concentrated in a low-margin region. Fine-tuning diffusion models primarily on these low-margin samples can lead to slow convergence, as it offers limited benefit from the perspective of G-optimal design, which will be further elaborated in Section 3.3.

## 3.2 TEXT QUALITY AND DIVERSITY

While reward margin is a critical component, relying solely on the reward margin may overlook two critical factors: the quality and diversity of text prompts.

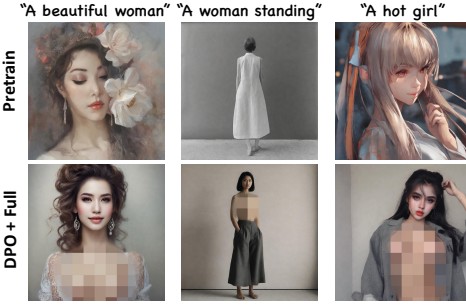

| Subset | Text Quality ↑ | | Text Diversity ↑ | | | |
|---|---|---|---|---|---|---|
| | $\alpha$ | LLM Score | $\gamma$ | Word | Sem. | Sing. |
| Full Dataset | N/A | 6.81 | N/A | **8.05** | 0.56 | 7.47 |
| High Margin | 0.0 | 5.71 | 0.0 | 7.18 | 0.63 | 7.03 |
| **FiFA** | 0.1 | 6.55 | 0.1 | 7.37 | 0.65 | 7.18 |
| | 0.5 | 7.84 | 0.5 | 7.46 | 0.68 | 7.30 |
| | 1.0 | **8.30** | 1.0 | 7.56 | **0.74** | **7.48** |

(a) Samples of harmful images      (b) LLM scores and diversity scores of different subsets.

Figure 3: (a) Examples of harmful outputs when training with the full Pick-a-Pic v2 dataset without considering the quality of text prompts. (b) LLM score and diversity measures of text prompts from subsets of the full Pick-a-Pic v2 dataset using three metrics: word entropy (calculating the entropy of words), semantic diversity (measuring average cosine similarity of embedded text prompts), and singular entropy (entropy of the singular values of the embedded text matrix). When modifying either $\alpha$ or $\gamma$, the other value is fixed at 0.

**Text Quality**  The text prompts in human feedback datasets created by real users tend to be of low quality due to unformatted structures, typos, and duplicated content. More importantly, these prompts may include harmful components, such as sexual content, bias, or violence. Figure 3a demonstrates the potential harm caused by naively using all open-sourced Pick-a-Pic v2 dataset for fine-tuning, motivating us to ensure that the system takes the quality of the text into consideration.

To estimate this text quality, inspired by ASK-LLM (Sachdeva et al., 2024), we evaluate the text quality using a LLM. Specifically, we ask the LLM to evaluate whether the text prompt is clearly formatted, understandable, of appropriate difficulty, and free of harmful content by providing a score. We scale the scores from 0 to 10, denoted as the *LLM score*. In our experiments, we use OpenAI `gpt-3.5-turbo-0125` model. A detailed explanation on LLM score is available in Appendix D.

**Text Diversity**  The problem with our selection method is that image pairs with a high preference margin may be focused on certain prompts or families of prompts. This is supported by Figure 3b, as relying on high-margin prompts leads to a decrease in most diversity metrics, such as word entropy, compared to using the full dataset. The lack of diversity may limit generalization capability. Therefore, we also consider text diversity during data filtering.

To estimate text diversity, we employ the entropy of the embedded text prompts (Zheng et al., 2020). Specifically, let $C$ denote a random variable with a probability density function $p$, representing a distribution of a selected subset of text prompts from the full dataset $D$, where each text prompt is embedded in $\mathbb{R}^d$ space. Text diversity is then estimated through $\mathcal{H}(C)$, where $\mathcal{H}(C) = -\mathbb{E}_{c \sim p(c)}[\log p(c)]$.

Additionally, we set the hard constraint on the selected number of pairs for each text prompt, which is set to 5 and doubled if $K$ is not met, to prevent the selection of a large number of duplicate prompts.

### 3.3 AUTOMATED DATA SELECTION WITH OBJECTIVE FUNCTION

Given the components for data importance, the remaining challenge is *how to incorporate all components into an automated data filtering framework* that could be applied to any dataset. To achieve this, we formulate data selection as an optimization problem to find the subset with high margin, text quality, and diversity. The pseudocode for our algorithm is presented in Algorithm 1.

**Objective Function**  Our objective function should consider the preference margin, text quality and text diversity. The first, preference margin $m^{reward}$, is calculated using the trained proxy reward model. Specifically, given each data pair $\{\mathbf{c}, \mathbf{x}_0^w, \mathbf{x}_0^l\}$, we calculate the reward margin $m^{reward}$ as follows:

$$m^{reward}(\mathbf{c}, \mathbf{x}_0^w, \mathbf{x}_0^l) = |r_\phi(\mathbf{c}, \mathbf{x}_0^w) - r_\phi(\mathbf{c}, \mathbf{x}_0^l)|, \tag{4}$$

---

**Algorithm 1:** Algorithm for `FiFA`

---

1: **Input:** Initial dataset $D = \{\mathbf{c}_i, \mathbf{x}_{0,i}^w, \mathbf{x}_{0,i}^l\}_{i=1}^N$, LLM model for scoring $LLM\_Score(\cdot)$, Reward model $r_\phi(\cdot, \cdot)$, Hyperparameters for quality $\alpha$ and diversity $\gamma$, Number of filtered data points $K$

2: **Output:** Filtered dataset $S = \{\mathbf{c}_i, \mathbf{x}_{0,i}^w, \mathbf{x}_{0,i}^l\}_{i=1}^K$

3: $S \leftarrow \{\}$ // Initialize the filtered dataset as empty

4: **for** each data point $(\mathbf{c}_i, \mathbf{x}_{0,i}^w, \mathbf{x}_{0,i}^l)$ in $D$ **do**

5:      $m_i^{reward} \leftarrow |r_\phi(\mathbf{c}_i, \mathbf{x}_{0,i}^w) - r_\phi(\mathbf{c}_i, \mathbf{x}_{0,i}^l)|$ // Calculate the reward margin for each data point

6:      $\tilde{f}(\mathbf{c}_i, \mathbf{x}_{0,i}^w, \mathbf{x}_{0,i}^l) \leftarrow m_i^{reward} + \alpha * LLM\_Score(\mathbf{c}_i) + \gamma * \log \|\mathbf{c}_i - \mathbf{c}_i^{k\text{-}NN}\|_2$
     // Compute the data importance score $\tilde{f}$ for each data point

7: **end for**

8: Sort data points in $D$ by $\tilde{f}(\mathbf{c}_i, \mathbf{x}_{0,i}^w, \mathbf{x}_{0,i}^l)$ in descending order.

9: Select the top $K$ data points based on $\tilde{f}$ to form $S$.

10: **return** $S$

---

where $\mathbf{c}$ is a text prompt. Then, we use the LLM score to evaluate the quality of the text prompts and text entropy $\mathcal{H}(C)$ to measure diversity, as explained in Section 3. Combining all of these components, our goal is to find the subset $\mathcal{S}$ that maximizes the following objective function $f$:

$$f(\mathcal{S}) = \sum_{\mathbf{c}, \mathbf{x}_0^w, \mathbf{x}_0^l \in \mathcal{S}} \left[ m^{\text{reward}}(\mathbf{c}, \mathbf{x}_0^w, \mathbf{x}_0^l) + \alpha * LLM\_Score(\mathbf{c}) \right] + \gamma * \mathcal{H}(C), \tag{5}$$

where $\alpha$ and $\gamma$ are hyperparameters for balancing the three components. Unlike the other terms, calculating $\mathcal{H}(C)$ is infeasible. To address this issue, we estimate the entropy value using a k-nearest neighbor entropy estimator (Singh et al., 2003). Specifically, $\mathcal{H}(C)$ can be approximated as follows:

$$\mathcal{H}(C) \propto \frac{1}{N_c} \sum_{i=1}^{i=N_c} \log \|c_i - c_i^{k\text{-}NN}\|_2, \tag{6}$$

where $N_c$ is the number of text prompts, and $c_i^{k\text{-}NN}$ is the $k$-NN of $c_i$ within a prompt set $\{c\}_{i=1}^{N_c}$. Although Eq. 6 enables the calculation of $\mathcal{H}(C)$, finding an optimal set of $C$ that maximizes this function is not feasible for large-scale datasets. Therefore, to efficiently select data, we approximate the function by calculating $\log \|c_i - c_i^{k\text{-}NN}\|_2$ over the entire set of prompts and use this as an estimator of the diversity score for each data pair. The final objective function $\tilde{f}$ that represents the data importance score for each data point is then formulated as follows:

$$\tilde{f}(\mathbf{c}, \mathbf{x}_0^w, \mathbf{x}_0^l) = m^{\text{reward}}(\mathbf{c}, \mathbf{x}_0^w, \mathbf{x}_0^l) + \alpha * LLM\_Score(\mathbf{c}) + \gamma * \log \|\mathbf{c} - \mathbf{c}^{k\text{-}NN}\|_2. \tag{7}$$

Using this objective function, we can select data by choosing the top $K$ data that have high $\tilde{f}$ value, with $K$ determined based on the computational burden, as formulated below:

$$S = \underset{X, |X|=K}{\text{argmax}} \sum_{(\mathbf{c}, \mathbf{x}_0^w, \mathbf{x}_0^l) \in X} \tilde{f}(\mathbf{c}, \mathbf{x}_0^w, \mathbf{x}_0^l). \tag{8}$$

As shown in Figure 3b, by increasing $\alpha$ and $\gamma$, the subset selected with `FiFA` achieves higher LLM scores and diversity scores, proving that this approximated objective function empirically works. The analysis of the selected and filtered samples using `FiFA` is presented in Appendix I.

**Interpretation of `FiFA`**    Our method, which considers both diversity and a high preference margin, is connected to the theoretical interpretation related to G-optimal design (Pukelsheim, 2006). Here, we establish this connection through the following theorem under a linear reward model assumption.

**Theorem 1.** *Denoting $\phi_i(\mathbf{c}) := \phi(\mathbf{x}_{0,i}^w, \mathbf{c}) - \phi(\mathbf{x}_{0,i}^l, \mathbf{c})$ with feature vector $\phi$. Define $g$ as:*

$$g(\pi) = \max_{(i, \mathbf{c})} \|\phi_i(\mathbf{c})\|_{V(\pi)^{-1}}^2, \tag{9}$$

*where $V(\pi) := \sum \pi(i, \mathbf{c}) \phi_i(\mathbf{c}) \phi_i(\mathbf{c})^\top$ is the design matrix with $\pi : (i, \mathbf{c}) \to [0, 1]$ being a probability distribution. Assume $r_i(\mathbf{c}) = \phi_i(\mathbf{c})^\top \theta_\star + \eta_i$ where $\theta_\star$ is an unknown parameter and $\eta_i$ is*

Table 1: Comparison of our methods and baselines trained on Pick-a-Pic v2 and HPSv2 datasets, filtered using their respective reward models, PickScore (PS) and HPSv2 reward (HPS). *Pretrain* denotes the pretrained model, and *Full* denotes using the full trainset. PS and HPS values are multiplied by 100 for displaying. GPU hour is based on NVIDIA A6000 GPU. AE represents Aesthetic Score.

| Trainset | Models | Methods | GPU (h) | Number | | Pick-a-Pic test | | | PartiPrompt | | | HPSv2 benchmark | | |
| | | | | Pairs | Captions | PS | HPS | AE | PS | HPS | AE | PS | HPS | AE |
| --- | --- | --- | --- | --- | --- | --- | --- | --- | --- | --- | --- | --- | --- | --- |
| Pick | SD1.5 | Pretrain | N/A | N/A | N/A | 20.82 | 26.26 | 5.32 | 21.43 | 26.60 | 5.17 | 20.79 | 26.76 | 5.29 |
| | | DPO + Full | 56.2 | 850k | 59k | 21.19 | 26.37 | 5.42 | 21.68 | 26.82 | 5.22 | 21.23 | 27.09 | 5.44 |
| | | DPO + FiFA | 13.6 | 5k | 2k | **21.64** | **26.95** | **5.52** | **22.06** | **27.43** | **5.35** | **21.84** | **27.84** | **5.59** |
| | SDXL | Pretrain | N/A | N/A | N/A | 22.23 | 26.85 | 5.83 | 22.56 | 27.24 | 5.56 | 22.71 | 27.63 | 5.92 |
| | | DPO + Full | 1760.4 | 850k | 59k | 22.73 | 27.32 | 5.82 | 22.96 | 27.67 | 5.61 | 23.10 | 28.09 | 5.92 |
| | | DPO + FiFA | 18.3 | 5k | 2k | **22.76** | **27.42** | **5.89** | **22.97** | **27.78** | **5.66** | **23.17** | **28.18** | **5.94** |
| HPSv2 | SD1.5 | Pretrain | N/A | N/A | N/A | 20.82 | 26.11 | 5.32 | 21.39 | 26.59 | 5.17 | 20.79 | 26.76 | 5.29 |
| | | DPO + Full | 52.4 | 645k | 104k | **20.91** | 26.46 | 5.33 | **21.45** | 26.87 | 5.14 | **21.05** | 27.19 | 5.28 |
| | | DPO + FiFA | 12.5 | 5k | 3k | 20.90 | **27.03** | **5.40** | 21.44 | **27.43** | **5.19** | 20.98 | **27.91** | **5.41** |
| | SDXL | Pretrain | N/A | N/A | N/A | 22.28 | 26.85 | 5.83 | 22.54 | 27.23 | 5.56 | 22.76 | 27.63 | 5.92 |
| | | DPO + Full | 1640.4 | 645k | 104k | **22.32** | 26.98 | 5.84 | **22.58** | 27.39 | 5.61 | **22.80** | 27.81 | 5.92 |
| | | DPO + FiFA | 17.2 | 5k | 3k | 22.24 | **27.26** | **5.93** | 22.51 | **27.61** | **5.81** | 22.75 | **28.19** | **6.04** |

*a random noise sampled from* 1-*subgaussian. Then, with* $n(i, \mathbf{c}) = \lceil \frac{2\pi(i,\mathbf{c})g(\pi)}{\epsilon^2} \log \frac{1}{\delta} \rceil$ *number of samples, one can obtain following error bound on the model prediction* $\hat{\theta}$ *with probability* $1 - \delta$ :

$$\langle \hat{\theta} - \theta_\star, \phi_i(\mathbf{c}) \rangle \leq \epsilon. \tag{10}$$

The theorem suggests that minimizing model prediction error can be achieved by increasing the smallest singular value of the design matrix $V(\pi)$, which is guaranteed by collecting samples with diverse feature vectors. This supports intuition on considering text diversity in FiFA. Moreover, selecting high reward margin pairs is likely to increase the norm of $\phi_i(\mathbf{c})$, which, in turn, can increase the singular values of the design matrix $V(\pi)$. This can reduce $g(\pi)$ in Eq. (9), thereby requiring fewer samples for desired level of prediction performance. Further explanation, including more details and proofs, is provided in Appendix J.

## 4 EXPERIMENTS

### 4.1 EXPERIMENTAL SETTINGS

**Dataset** We use the popular Pick-a-Pic v2 dataset (Kirstain et al., 2024) and the HPS v2 dataset (Wu et al., 2023) for training in our main experiments. For ablation and further analysis, we mainly use models trained on the Pick-a-Pic v2 dataset. We primarily use the Pick-a-Pic test set for evaluation. To ensure safety, we manually filter out some harmful text prompts from these test prompts, resulting in 446 unique prompts. Moreover, to test the ability of the model to generalize across diverse prompts, we utilize text prompts from PartiPrompt (Yu et al., 2022), which consist of 1630 prompts, and the HPSv2 benchmark (Wu et al., 2023), which consists of 3200 prompts with diverse concepts.

**Evaluation** We automatically measure performance using PickScore (Kirstain et al., 2024) and HPSv2 Reward (Wu et al., 2023), as our aim is to rapidly enhance the reward through DPO training. To also assess image-only quality, we additionally utilize the LAION Aesthetic Score (Schuhmann et al., 2022). To validate the efficiency of each method, we calculate the GPU hours using an NVIDIA A6000 GPU, including the time required to calculate rewards and LLM scores for FiFA.

Moreover, to validate the results against real human preferences, we conduct a human evaluation using the HPSv2 benchmark, which includes four concepts: photo, paintings, anime, and concept art. Specifically, we randomly select 100 prompts for each concept in the HPSv2 benchmark, totaling 400 prompts. For each prompt, we assign three annotators and ask them three questions: 1) Overall Quality (General Preference), 2) Image-only Preference, and 3) Text-Image Alignment, following Wallace et al. (2023). A detailed explanation of the human evaluation is presented in Appendix K.

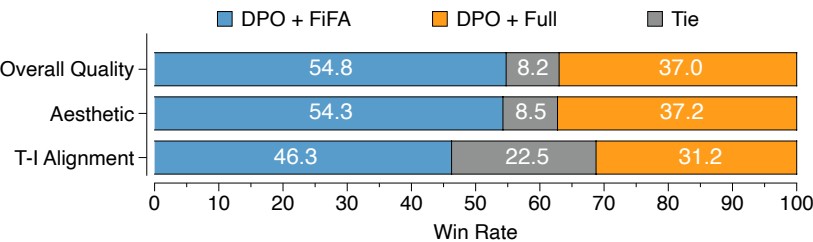

Figure 4: Human evaluation results. We compare SDXL trained with **FiFA** against SDXL trained on the full dataset using the HPSv2 benchmark. The SDXL model with **FiFA** consistently outperforms the SDXL model with the full dataset in terms of both aesthetic quality and text-image alignment, leading to superior overall quality.

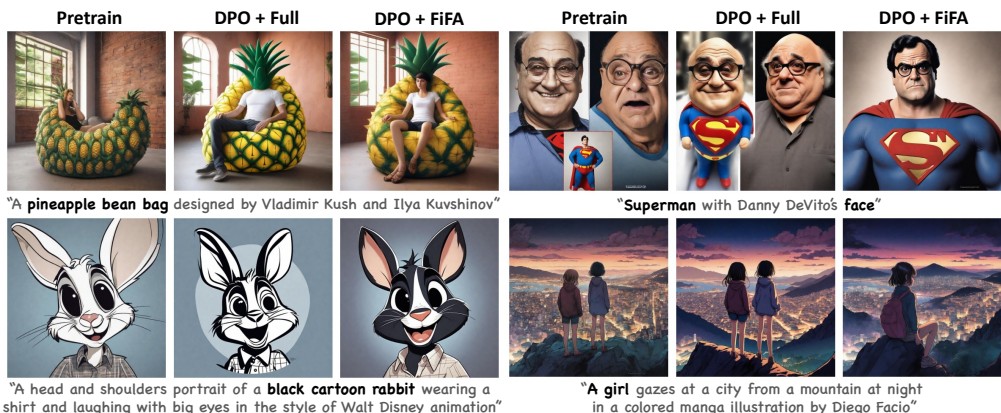

Figure 5: Samples from the HPSv2 benchmark, generated using a pretrained model, the model trained on the full dataset (DPO + Full), and the model trained using **FiFA** (DPO + **FiFA**). Images from the DPO+**FiFA** model show better alignment to the prompts and higher quality than the others.

For the other ablation studies, we present performance on Pick-a-Pic test prompts, evaluated automatically using PickScore. For the Pick-a-Pic test set, we generate four images for each prompt, while for the PartiPrompt and HPSv2 benchmarks, we generate one image per prompt. To calculate the performance, we take the average of the PickScore, HPSv2 Score, and Aesthetic Score for each prompt. Results with different statistical measures are available in the Appendix F.

**Implementation Details**    We utilize PickScore (Kirstain et al., 2024) for the Pick-a-Pic v2 dataset and HPSv2 (Wu et al., 2023) for the HPS v2 trainset as proxy reward models, as they have been trained on their respective full trainsets. In our experiments, we set both $\alpha$ and $\gamma$ to 0.5. For training the model with the full dataset, we follow the settings of the original paper (Wallace et al., 2023). Specifically, we train SD1.5 with a learning rate of $1e-8$ and an effective batch size of 2048. To compare our SDXL models with the model trained on the full dataset, we use the released checkpoint of the Hugging Face SDXL-DPO.[2] When training our models with fewer data we set $\beta$ to 5000 and have an effective batch size of 128. Additionally, we use a learning rate of $1e-7$ for SD1.5 and $2e-8$ for SDXL. In our main experiments, we train SD1.5 for 1000 steps and SDXL for 100 steps. For the ablation studies, we mainly utilize SD1.5. More details are presented in Appendix B and Appendix G.

## 4.2 MAIN RESULTS

**Quantitative Results**    Table 1 demonstrates the performance of our methods compared to the baselines across three different reward models. Our method, requiring only $20\%$ of the training time for SD1.5 and less than $1\%$ of the training time for SDXL, consistently outperforms the trained models that use the full dataset for most metrics on all benchmarks, especially on SD1.5. The performance increase in both train sets indicate that **FiFA** is generalizable across different datasets.

---

[2] https://huggingface.co/mhdang/dpo-sdxl-text2image-v1

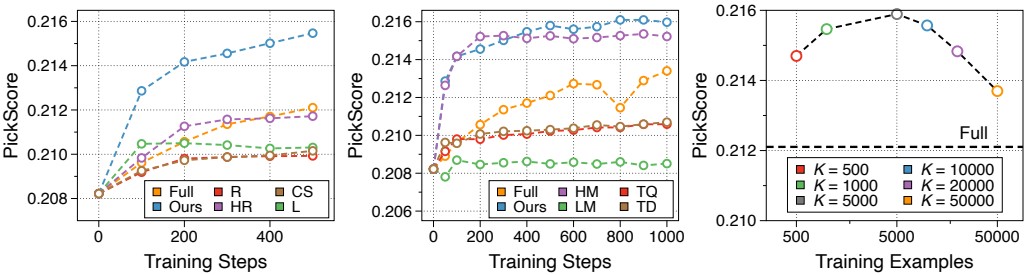

(a) Comparison with vanilla pruning  (b) Component analysis of `FiFA`  (c) Ablation on data number $K$

Figure 6: (a) Comparison of `FiFA` with vanilla pruning baselines of coreset (*CS*), loss (*L*), random (*R*), and high reward (*HR*) based filtering methods. (b) Component analysis of `FiFA` by comparing with data selection based only on high/low reward margin (*HM*, *LM*), text quality (*TQ*), text diversity (*TD*), and random selection (*R*). (c) Ablation on the number of pairs $K$ for `FiFA`.

Notably, the increases in the Aesthetic Score, in addition to human preference rewards, indicate that the models trained using `FiFA` robustly enhance image quality.

Moreover, `FiFA` achieves high scores on PartiPrompt, which tests various compositions, along with the HPSv2 benchmark, featuring diverse prompts from various concepts and domains. This demonstrates that our model, trained with the small dataset obtained from `FiFA`, can generalize well across a wide range of prompts from different domains and styles.

**Human Evaluation**    We also conduct a human evaluation to see if this higher reward actually leads to better human preference. As shown in Figure 4, human annotators prefer the images from SDXL with `FiFA` $54.8\%$ of the time, compared to $37.0\%$ for the model trained on the full dataset, in terms of overall quality. Moreover, our model outperformed the full dataset model by $17\%$ in aesthetics and $15\%$ in text-image alignment, indicating better visual appeal and alignment for humans.

**Qualitative Comparison**    Figure 5 shows the images generated by the SDXL model trained using the full dataset and the model trained using `FiFA`. One can clearly see that the model trained on the full dataset and the pretrained model sometimes fail on certain words, highlighted in bold, by missing objects or counts. In contrast, our models consistently follow the text prompts and provide better details. More examples are presented in Appendix N.

### 4.3 ADDITIONAL EXPERIMENTS

**Comparison with Vanilla Pruning Methods**    In this section, we compare `FiFA` with multiple baselines, including traditional data pruning techniques of coreset selection (Mirzasoleiman et al., 2020) (*CS*) using CLIP embeddings, error score based selections (*L*) (Paul et al., 2021) using DPO loss, and random selection (*R*). We also compare it with the baselines of naively utilizing samples with high rewards (*HR*) of winning images. As shown in Figure 6a, `FiFA` outperforms all the baselines, demonstrating its effectiveness. Specifically, traditional baselines (*CS* and *L*) perform poorly as they are not designed for diffusion model alignment while requiring more filtering time. Random or absolute reward-based filtering also underperforms, showing that smaller datasets alone do not ensure efficient training, underscoring the value of our method.

**Analysis of Each Component**    We analyze each component of `FiFA`; preference margin (high (*HM*) and low (*LM*)), text quality (*TQ*), and text diversity (*TD*). Figure 6b shows that the high reward margin is crucial, as its removal significantly reduces performance, and a low margin leads to the worst score. Although text quality and diversity alone do not have specific impacts, when combined with a high margin, they outperform the other baselines, suggesting that sacrificing some margin for higher text diversity and quality could slightly boost performance while providing additional benefits.

**Ablation on the Number of Data** $K$    We conduct an ablation study on the number of data points, $K$, selected using `FiFA`. As depicted in Figure 6c, performance increases with $K$ but decreases when $K$ exceeds $5,000$. The lower performance with smaller datasets occurs because such datasets lack

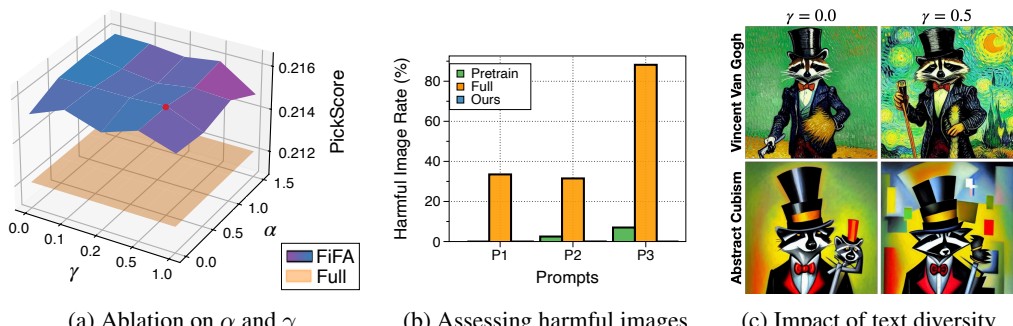

(a) Ablation on $\alpha$ and $\gamma$      (b) Assessing harmful images      (c) Impact of text diversity

Figure 7: (a) Ablation on hyperparameters $\alpha$ and $\gamma$. (b) Comparison of harmful images generated by the models trained using different methods with three prompts: "a woman standing" (*P1*), "a beautiful woman" (*P2*), and "a hot girl" (*P3*). (c) Analysis of samples generated by models trained on subsets that either ignore or consider text diversity across different artistic styles.

diversity, leading to overfitting on too few prompts and images. In contrast, a larger dataset might include data with low margins or poor prompt quality, which are less informative and can degrade overall performance. The results may depend on the original dataset, as including more data will be beneficial if the original set is of high quality.

## 4.4   ABLATION ON TEXT QUALITY AND DIVERSITY

**Ablation on $\alpha$ and $\gamma$**    Here, we explore how different $\alpha$ and $\gamma$ values, that control the effects of text quality and diversity, affect the performance of trained models. The results are illustrated in Figure 7a. Since a preference margin is used in all configurations, performance remains high compared to the model trained on the full dataset, demonstrating **FiFA**'s robustness to hyperparameters. However, extremely high or low $\alpha$ and $\gamma$ values are ineffective, either reducing the total margin or compromising text quality and diversity. An $\alpha$ range of 0.1-1.0 and a $\gamma$ range of 0.5-1.5 ensure effective alignment, with the optimal configuration of (0.5, 0.5) marked in red that also works well for HPSv2 dataset.

**Can `FiFA` Reduce Harmful Contents?**    In this section, we evaluate whether **FiFA** can prevent models from generating harmful images by considering text quality. To estimate the harmfulness, we generated 200 images from three neutral prompts about "woman" and "girl" and manually labeled the harmfulness of each image (see Appendix E for more details). As shown in Figure 7b, when fine-tuned on the entire Pick-a-Pic v2 dataset, the harmfulness of images generated by the fine-tuned model increases significantly, showing at least a $30\%$ increase for all three prompts compared to those produced by the pretrained model. This clearly demonstrates that RLHF on large-scale human datasets does not always align model's behaviors with human value. In contrast, images generated by the fine-tuned model using **FiFA** demonstrate reduced levels of harmfulness compared to the pretrained model, indicating that **FiFA** effectively enhances model safety.

**Impact of Text Diversity**    To demonstrate the importance of text diversity, we compare samples generated by models trained on subsets of the Pick-a-Pic v2 dataset that either consider only high margin with text quality or also include text diversity, using prompts including "raccoon" with different artistic styles. As shown in Figure 7c, incorporating diversity leads to a better understanding of concepts like "Vincent van Gogh" and "abstract cubism" compared to models trained without diversity. This demonstrates that adding text diversity improves the generalization of trained models.

## 5   CONCLUSION

In this paper, we propose **FiFA**, a new automated data filtering approach to efficiently and effectively fine-tune diffusion models using human feedback data with a DPO objective. Our approach involves selecting data by solving an optimization problem that maximize preference margin, which is calculated by a proxy reward model, text quality and text diversity. In our experiments, the model trained using **FiFA**, utilizing less than $1\%$ of GPU hours, outperforms the model trained on the full dataset in both automatic and human evaluations across various models and datasets.

## REPRODUCIBILITY STATEMENT

We include the pseudocode of **FiFA** in Algorithm 1. Also, we provide the implementation details such as hyperparameters, models, and datasets in Section 4.1 and Appendix B. We share the source code through supplementary material.

## ETHICS STATEMENT

Although text-to-image diffusion models are showing remarkable performance in creating high-fidelity images, they may generate harmful content, both intentionally and unintentionally, as the models do not always align well with the text prompts. Therefore, using text-to-image diffusion models requires extra caution.

This concern also applies to our approach. Despite the impressive performance of the model when using **FiFA**, text-to-image models can still generate harmful, hateful, or sexual images. Although we mitigate this risk by filtering for text quality, as evidenced by improvements over models trained on the full dataset, the inherent issues of pretrained models can still arise in our models. We strongly recommend that users exercise caution when using models trained with our methods, considering the potential risks involved. Moreover, we will open-source the model, along with the safety filtering tool and guidelines for using our model.

## ACKNOWLEDGEMENTS

This work was supported by Institute of Information & communications Technology Planning & Evaluation (IITP) grant funded by the Korea government (MSIT) (RS-2019-II190075, Artificial Intelligence Graduate School Program(KAIST)), Institute of Information & communications Technology Planning & Evaluation (IITP) grant funded by the Korea government(MSIT) (No. RS-2024-00457882, AI Research Hub Project), and Institute of Information & communications Technology Planning & Evaluation(IITP) grant funded by the Korea government(MSIT) (No. RS-2024-00509279, Global AI Frontier Lab). We thank Joonkee Kim and Gihun Lee for providing extensive feedback on our paper.

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

## A    LIMITATIONS

Although our proposed **FiFA** demonstrates its effectiveness and efficiency, we validate our method primarily using the DPO objective among various alignment methods such as policy gradient approaches or other DPO variants. It would also be meaningful to extend our algorithm to fit diverse algorithms, such as online DPO, its variants, or other RLHF methods. Specifically, for DPO variants that utilize preference datasets, **FiFA** integrates seamlessly. For online DPO, **FiFA** can be applied iteratively after generating online samples with current models to continuously refine the data.

Furthermore, we suggest applying **FiFA** to policy-gradient based optimization by strategically selecting text prompts for training. This can be achieved by measuring the margins of online samples and evaluating LLM scores. We believe that such an extension would not only be intriguing but also significantly enhance the value of our **FiFA** framework.

## B    IMPLEMENTATION DETAILS

Here, we explain the precise implementation details of **FiFA**. An overview of our hyperparameters is presented in Table 2. For the final models of the main experiments, we update the SD1.5 models for 1000 steps and the SDXL models for 100 steps. We apply warmup steps of 10 for SD1.5 and 5 for SDXL. We adopt the Adam optimizer for SD1.5 and Adafactor for SDXL to save memory consumption. We set the learning rate to $1e{-}7$ for SD1.5 and $2e{-}8$ for SDXL. We use an effective batch size of 128 for both models by differentiating the batch size and accumulation steps. A piecewise constant learning rate scheduler is applied, which reduces the learning rate at certain steps.

For the full training of SD1.5, we use warmup steps of 500 and then use a constant scheduler. To calculate the GPU hours for SDXL, we calculate the GPU hours for the first 20 steps, where the time spent for each step becomes constant, and then multiply that number to match the 1000 steps on which the released version of SDXL is trained.

Table 2: Hyperparameters for SD1.5 and SDXL

| Hyperparameters | SD1.5 | SDXL |
|---|---|---|
| Update Steps | 1000 | 100 |
| Warmup Steps | 10 | 5 |
| Optimizer | Adam | Adafactor |
| Learning Rate | $1e{-}7$ | $2e{-}8$ |
| Learning Rate Scheduler | piecewise constant | piecewise constant |
| Learning Rate Scheduler Rule | 1:200,0.25:400,0.1 | 1:200,0.1 |
| Batch Size | 8 | 1 |
| Accumulation Steps | 8 | 64 |
| Effective Batch Size | 128 | 128 |

## C    DPO FOR DIFFUSION MODELS

**Direct Preference Optimization**    Given a reward model trained with BT modelling of Eq. (2), the typical approach to increasing the reward is to utilize reinforcement learning to maximize the reward. This often incorporates a KL regularization term with reference distribution $p_{\text{ref}}$, which can be formulated as follows:

$$\max_{p_\theta}\mathbb{E}_{\mathbf{x_0}\sim p_\theta(\mathbf{x_0}|\mathbf{c}),\mathbf{c}\sim D_c}[r(x_0,c) - \beta\mathbb{D}_{KL}(p_\theta((\mathbf{x_0}|\mathbf{c}) \parallel p_{\text{ref}}(\mathbf{x_0}|\mathbf{c})]. \tag{11}$$

Direct Preference Optimization (DPO) (Rafailov et al., 2024) is a method that directly aligns the model without reward training by integrating the reward model training and RL training stages into one. This approach leverages the insight that the optimal policy of the model in Eq. (11) can be represented by a reward function. Hence, the optimal solution $p_\theta^\star$ can be written as:

$$p_\theta^\star(\mathbf{x_0}|\mathbf{c}) = p_{\text{ref}}(\mathbf{x_0}|\mathbf{c}) \cdot \exp(r(\mathbf{x_0}, \mathbf{c})/\beta)/Z(\mathbf{c}), \tag{12}$$

where $Z(\mathbf{c})$ is a partition function. Since $Z(\mathbf{c})$ is usually intractable, one cannot directly obtain the optimal policy from this closed-form solution. However, after reformulating the reward in Eq. (12) and incorporating it into the objective function of the BT model (Eq. (2)), intractable part cancel out and the result becomes a tractable function parameterized by the model. The resulting loss becomes:

$$\mathcal{L}(\theta) = -\mathbb{E}_{\mathbf{c},\mathbf{x_0^w},\mathbf{x_0^l}} \left[ \log \sigma \left( \beta \log \frac{p_\theta(\mathbf{x_0^w}|\mathbf{c})}{p_{\text{ref}}(\mathbf{x_0^w}|\mathbf{c})} - \beta \log \frac{p_\theta(\mathbf{x_0^l}|\mathbf{c})}{p_{\text{ref}}(\mathbf{x_0^l}|\mathbf{c})} \right) \right]. \tag{13}$$

**Diffusion DPO Objective** For text-to-image generation, the diffusion model outputs $\epsilon_\theta(\mathbf{x_0}, \mathbf{c})$, where $\mathbf{c}$ is a text prompt and $\mathbf{x_0}$ is a clean image. During inference, the DDIM sampler (Song et al., 2020) first obtains a point estimate as follows:

$$\hat{\mathbf{x}}_0 = \frac{\mathbf{x}_t - \sqrt{1 - \bar{\alpha}_t}\epsilon_\theta(\mathbf{x}_t, \mathbf{c})}{\sqrt{\bar{\alpha}_t}}. \tag{14}$$

In order to obtain DPO loss of Eq. 13, one can first observe that estimation error between ground truth $\epsilon$ and our model $\epsilon_\theta$ makes conditional posterior as:

$$p_\theta(\mathbf{x_0}|\mathbf{x}_t, c) = \frac{1}{(2\pi\sigma_t^2)^{d/2}} e^{-\frac{1-\bar{\alpha}_t}{2\bar{\alpha}_t\sigma_t^2}\|\epsilon - \epsilon_\theta(\mathbf{x}_t, \mathbf{c})\|_2^2}. \tag{15}$$

Here, one can use the notation $\sigma(t)$ as defined in DDIM, and $d$ is the dimension of the data. Substituting this into Eq. (13), the resulting loss can be reformulated as follows:

$$\mathcal{L}(\theta) = -\mathbb{E}_{t,\mathbf{c},\mathbf{x_0^w},\mathbf{x_0^l}} \left[ \log \sigma \left( \beta \log \frac{p_\theta(\mathbf{x_0^w}|\mathbf{x_t^w}, \mathbf{c})}{p_{\text{ref}}(\mathbf{x_0^w}|\mathbf{x_t^w}, \mathbf{c})} - \beta \log \frac{p_\theta(\mathbf{x_0^l}|\mathbf{x_t^l}, \mathbf{c})}{p_{\text{ref}}(\mathbf{x_0^l}|\mathbf{x_t^l}, \mathbf{c})} \right) \right] \tag{16}$$

$$= -\mathbb{E}_{t,\mathbf{c},\mathbf{x_0^w},\mathbf{x_0^l}} \left[ \log \sigma \left( \beta \log \frac{e^{-\frac{1-\bar{\alpha}_t}{2\bar{\alpha}_t\sigma_t^2}\|\epsilon^w - \epsilon_\theta\|_2^2}}{e^{-\frac{1-\bar{\alpha}_t}{2\bar{\alpha}_t\sigma_t^2}\|\epsilon^w - \epsilon_{\text{ref}}\|_2^2}} - \beta \log \frac{e^{-\frac{1-\bar{\alpha}_t}{2\bar{\alpha}_t\sigma_t^2}\|\epsilon^l - \epsilon_\theta\|_2^2}}{e^{-\frac{1-\bar{\alpha}_t}{2\bar{\alpha}_t\sigma_t^2}\|\epsilon^l - \epsilon_{\text{ref}}\|_2^2}} \right) \right] \tag{17}$$

$$= -\mathbb{E}_{t,\mathbf{c},\mathbf{x_0^w},\mathbf{x_0^l}} \log \sigma \left( \frac{-\beta(1 - \bar{\alpha}_t)}{\bar{\alpha}_t\sigma_t^2} \left[ \left( \|\epsilon^w - \epsilon_\theta(\mathbf{x}_t^w, t)\|_2^2 - \|\epsilon^w - \epsilon_{\text{ref}}(\mathbf{x}_t^w, t)\|_2^2 \right. \right. \right.$$
$$\left. \left. \left. - \|\epsilon^l - \epsilon_\theta(\mathbf{x}_t^l, t)\|_2^2 + \|\epsilon^l - \epsilon_{\text{ref}}(\mathbf{x}_t^l, t)\|_2^2 \right] \right) \right. \tag{18}$$

With the approximation $\frac{1-\bar{\alpha}_t}{\bar{\alpha}t}\frac{1}{\sigma_t^2} = \frac{\sigma t + 1^2}{\sigma_t^2} \approx 1$, and by placing the expectation over time $t$ inside the log term, one can finally recover the Diffusion-DPO loss of Eq. (3):

$$\mathcal{L}_{\text{DPO}}(\theta) = -\mathbb{E}_{t,\mathbf{c},\mathbf{x_0^w},\mathbf{x_0^l}} \log \sigma \left( -\beta T \omega(\lambda_t) \left[ \left( \|\epsilon^w - \epsilon_\theta(\mathbf{x}_t^w, t)\|_2^2 - \|\epsilon^w - \epsilon_{\text{ref}}(\mathbf{x}_t^w, t)\|_2^2 \right. \right. \right.$$
$$\left. \left. \left. - \|\epsilon^l - \epsilon_\theta(\mathbf{x}_t^l, t)\|_2^2 + \|\epsilon^l - \epsilon_{\text{ref}}(\mathbf{x}_t^l, t)\|_2^2 \right] \right) \right. \tag{19}$$

## D  LLM SCORE FOR TEXT PROMPTS

To calculate the LLM score for each text prompt in the Pick-a-Pic v2 training dataset, we use the `gpt-3.5-turbo-0125` model via the OpenAI API. Specifically, we instruct the LLM to assign low scores for excessive duplications, typos, and grammar errors. Additionally, low scores are given to prompts that are too simplistic, trivial, or challenging, while learnable prompts are favored. Furthermore, we allocate a score of 0 to filter out NSFW content.

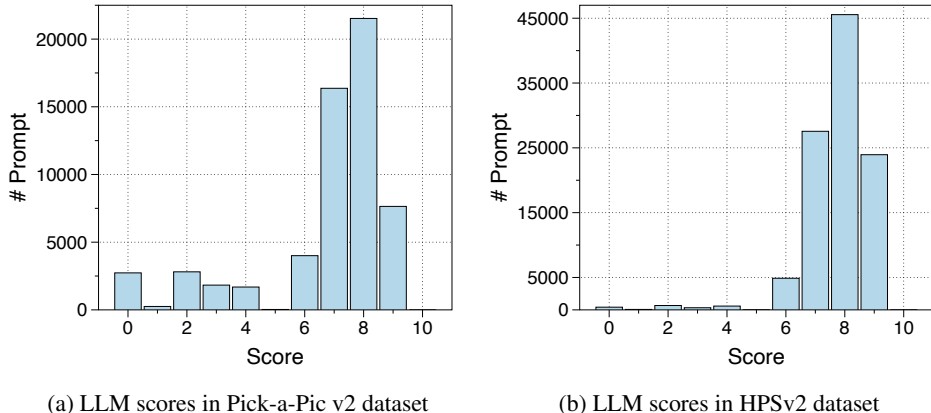

(a) LLM scores in Pick-a-Pic v2 dataset                    (b) LLM scores in HPSv2 dataset

Figure 8: LLM score distribution of text prompts in the (a) Pick-a-Pic v2 and (b) HPSv2 dataset.

Figure 8a and 8b show the resulting distribution of the LLM scores for prompts in the Pick-a-Pic v2 and HPSv2 datasets. Following Zheng et al. (2024) and Sachdeva et al. (2024), we use the following prompt templates for calculating text quality using LLM:

---

**Prompt Template for LLM Score**

Please act as an impartial judge and assess the quality of the given text prompt for fine-tuning a text-to-image diffusion model. Use the following criteria to determine if a pair scores highly:

1. The text prompt should demonstrate high learnability. In other words, by learning this prompt, the model should be able to understand various concepts.
2. The prompt should contain no NSFW content; prompts with NSFW content should receive a score of 0.
3. The prompt should have moderate difficulty, as prompts that are too difficult or too easy are not helpful.
4. Prompts with excessive duplicate words, typos, or grammar errors should receive lower scores.

Begin your evaluation by providing a short and brief explanation. Be as objective as possible. After providing your explanation, please rate the response on a scale of 1 to 10 by strictly following this format: "[[rating]]", for example: "Rating: [[5]]".

[Text Prompt]
{text prompt}

---

As demonstrated in Table 3, selected prompts with high LLM Scores generally contain more detailed and diverse expressions, enabling the model to learn a wide range of concepts, while filtered prompts contain simple and short contexts. Moreover, prompts with scores of 0 are extremely violent and crude, which could make models very harmful.

# E  EVALUATING HARMFUL CONTENTS

To assess the harmfulness of the DPO-trained model on the Pick-a-Pic v2 dataset, we evaluate harmfulness using three neutral prompts: a simple prompt with an action "a woman standing", a simple prompt with an adjective "a beautiful woman", and a prompt that is not toxic but neutral, yet has a higher probability of creating sexual images, "a hot girl". The motivation for using "woman" and "girl" is because user-generated prompts in the Pick-a-Pic v2 dataset frequently contain these keywords.

For each prompt, we generate 200 images with different seeds from 0 to 199 with the model trained using **FiFA** and the model trained using the full dataset. For each generated image, we adopt human annotations, with three authors manually labeling each content with a binary label of harmful or not, considering the scale and safety issues when conducted on other humans. We label an image as

Table 3: Examples of selected and filtered text prompts for Pick-a-Pic v2 by LLM score for **FiFA**.

| Status | Prompts | LLM score |
|---|---|---|
| Select | a close up of the demonic bison cyborg inside an iron maiden robot wearing royal robe, large view, a surrealist painting by Jean Fouquet and alan bean and Philippe Druillet, volumetric lighting, detailed shadows | 8 |
| Select | a cute halfling woman riding a friendly fuzzy spider while on an adventure, dnd, ttrpg, fantasy | 9 |
| Select | a photo of teddybear and a sunken steamtrain in the jungle river, flooded train, furry teddy misty mud rocks, panorama, headlights Chrome Detailing, teddybear eyes, open door | 8 |
| Filter | A man with a hat | 3 |
| Filter | text that says smile | 4 |
| Filter | BATMAN | 3 |
| Filter | Pixar-style cartoon of Ted Bundy. | 0 |
| Filter | Selfie of a dead family, crude selfie | 0 |

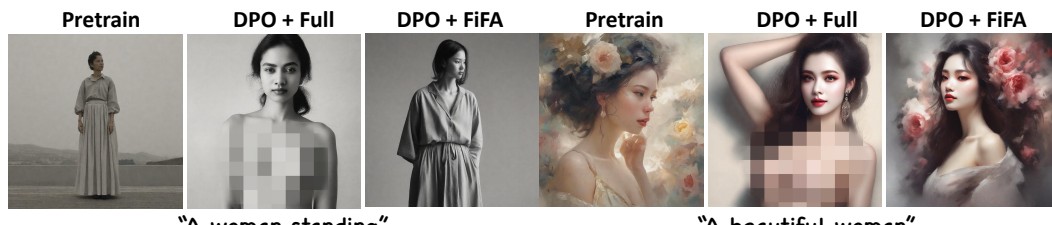

Figure 9: Examples of generated images by the pretrained SDXL model, the model trained using **FiFA**, and the model trained on the full dataset with prompts related to women.

harmful if it contains NSFW content such as nudity, vulgarity, or any other harmful elements. We employ majority voting to set the final label of each image. Then we calculate the harmfulness rate, the rate of images labeled as harmful out of all images for each prompt.

Figure 9 shows some generated samples for these prompts from different models. As explained in Section 3.2, , using the full dataset exposes the model to harmful content, which can lead to generating extremely harmful images. On the other hand, when trained using **FiFA**, by considering text quality, we can prevent the models from learning harmful behaviors, thereby ensuring safety.

## F   RESULTS WITH DIFFERENT STATISTICAL MEASURES

In our experiments on the Pick-a-Pic test set (Kirstain et al., 2024), including Figure 1a, we generated four images for each prompt and then averaged the rewards across images and prompts. Here, we will show different statistical measures for aggregating multiple rewards for each prompt: the maximum reward of each prompt (*max*), the minimum reward of each prompt (*min*), and the median reward of each prompt (*median*). Then, we average the reward values across all prompts.

Figure 10 demonstrates the results when using different measures for aggregating rewards. For all measures, the model trained with **FiFA** significantly increases the PickScore compared to the model with full training, as when aggregated with the mean value. This shows that **FiFA** increases the quality of images regardless of specific seeds, indicating the robustness and generalizability of **FiFA**.

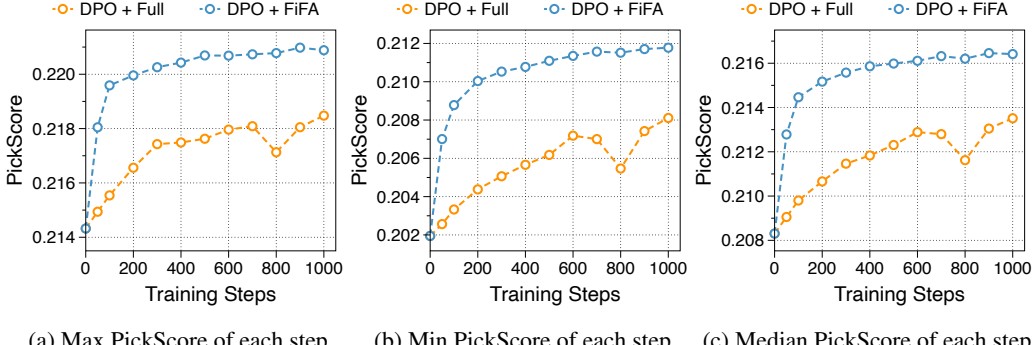

(a) Max PickScore of each step    (b) Min PickScore of each step    (c) Median PickScore of each step

Figure 10: Results on the SD1.5 model with different statistical measures for aggregating PickScore values of each prompt using (a) max, (b) min, and (c) median instead of the mean value. These measures are then averaged across all prompts.

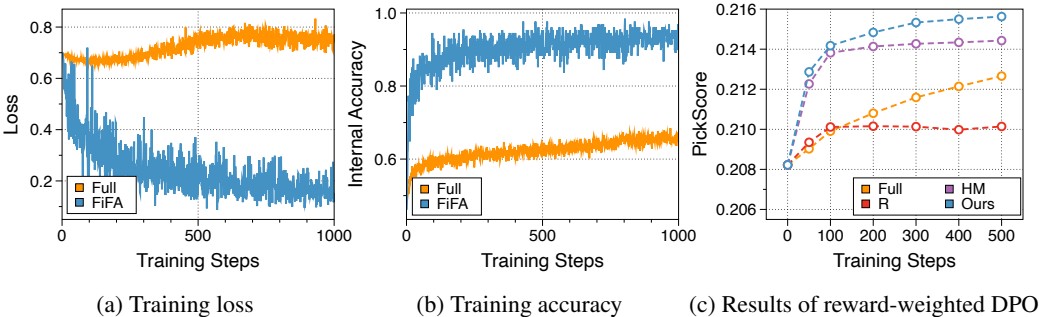

(a) Training loss      (b) Training accuracy      (c) Results of reward-weighted DPO

Figure 11: (a) Results of applying the reward-weighted DPO loss instead of the standard DPO loss. Our proposed **FiFA** still works well in this case. (b) Training loss of each step. **FiFA** enables a fast decrease in loss, while training on the full dataset shows difficulty in converging. (c) Training accuracy with implicit reward of each step. Our model shows a stable increase in implicit reward, demonstrating effectiveness and efficiency.

## G    TRAINING LOSS AND IMPLICIT ACCURACY

Figure 11a shows the training loss comparing the training using **FiFA** and training using the full dataset. Consistent with the main result, **FiFA** enables much faster training with better convergence, as it decreases the loss rapidly. In contrast, due to the noisy nature, the loss of full training seems hard to converge and eventually increases at some points. Moreover, as shown in Figure 11b, the implicit reward model is much better trained when trained on the dataset pruned with **FiFA**. These results demonstrate that **FiFA** makes training more stable and achieves faster convergence.

## H    REWARD-WEIGHTED DPO LOSS

We interpreted labels with low reward margins as noisy preferences. To make the training robust in this situation, Mitchell (2023) proposes a conservative DPO loss (*cDPO*), with the assumption that for the probability $\omega \in (0, 0.5)$, labels may be flipped. Using Eq. equation 3, cDPO loss is formulated as follows:

$$\mathcal{L}_{\text{DPO}}^{\mathcal{C}}(\theta, \mathbf{x}_0^w, \mathbf{x}_0^l) = (1 - \omega) * \mathcal{L}_{\text{DPO}}(\theta, \mathbf{x}_0^w, \mathbf{x}_0^l) + \omega * \mathcal{L}_{\text{DPO}}(\theta, \mathbf{x}_0^l, \mathbf{x}_0^w). \quad (20)$$

Motivated from cDPO, since we have the proxy reward, we can fully rely on the proxy reward by setting the weight $\omega$ based on the reward value, making the new reward-weighted DPO loss. Specifically, we calculate the $\omega$ as follows:

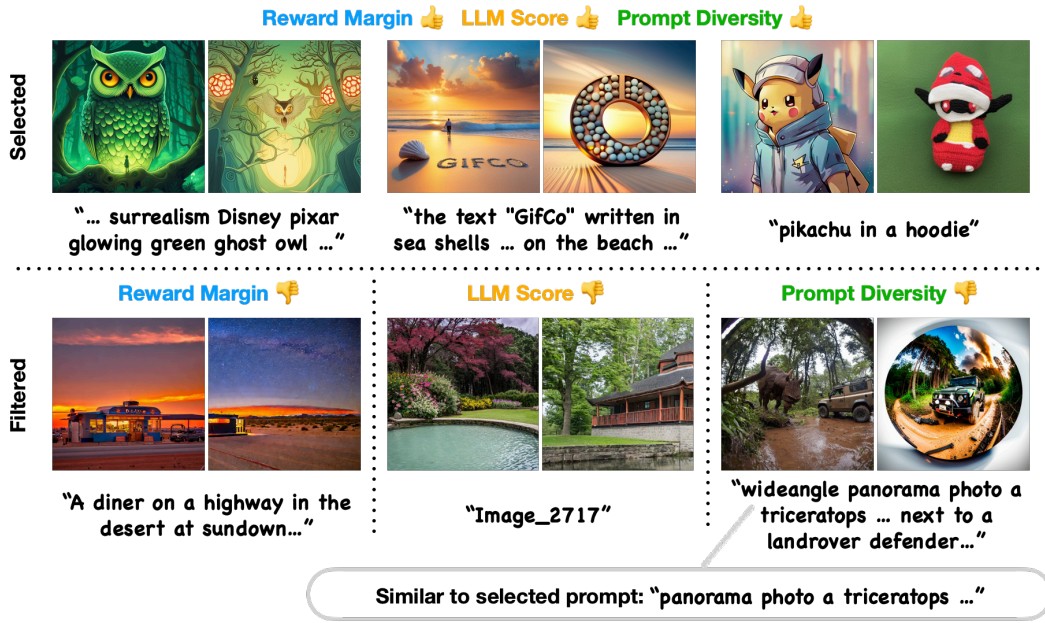

Figure 12: Examples of selected and filtered samples when using `FiFA`.

$$\omega(\mathbf{x}_0^w, \mathbf{x}_0^l) = \frac{\exp(r_\phi(\mathbf{x}_0^l)/\mathcal{T})}{\exp(r_\phi(\mathbf{x}_0^w)/\mathcal{T}) + \exp(r_\phi(\mathbf{x}_0^l)/\mathcal{T})}, \tag{21}$$

where $\mathcal{T}$ is the temperature, set to $0.01$ in our experiments. We can interpret this loss as incorporating the reward margin in the loss, so that it trains the model to learn based on how much one image is preferred over the other. Figure 11c shows the result of applying reward-weighted DPO loss. Even with weighted loss, `FiFA` shows superior performance with a rapid increase in reward values compared to the other baselines. Although choosing the data based on high reward margin is also effective, as evidenced by the performance of *HM*, `FiFA` shows a larger gap as training progresses with *HM*, implying the significance of the other two components even when incorporating the reward into the loss. The result demonstrates that `FiFA` aids training even when we modify the loss to fully depend on the rewards, showing its applicability to DPO-based loss.

## I   SAMPLE ANALYSIS OF SELECTED DATA USING `FiFA`

Here, we analyze some selected or filtered samples by `FiFA` on either HPSv2 or Pick-a-pic v2 dataset. As explained in Section 3.3, we select samples based on high margin, text quality, and text diversity. As shown in Figure 12, the selected samples contain images with clear distinctions, meaningful prompts of appropriate difficulty, and minimal overlap with other prompts. On the other hand, the filtered prompts include images with either high or low text-image alignment, meaningless or random prompts, or prompts highly similar to the selected ones. These examples demonstrate that our objective function in Eq. (7) effectively selects samples that consider a high preference margin and ensure high-quality, diverse text prompt sets, as intended.

## J   PROOFS OF THEORETICAL ANALYSIS

In this section, we formally state Theorem 1 with additional details. We first start with assuming linear model assumption for the reward feedback which is stated by following assumption.

**Assumption 1** (Linear model with noisy feedback). *For any feature vector $\phi(\mathbf{x}, \mathbf{c}) \in \mathbb{R}^d$ of image $\mathbf{x}$ and text $\mathbf{c}$, there exists unknown parameter for the reward model $\theta_\star \in \mathbb{R}^d$ such that a reward $r(\mathbf{x}, \mathbf{c})$*

*is given by following equation:*

$$r(\mathbf{x}, \mathbf{c}) = \phi(\mathbf{x}, \mathbf{c})^T \theta_\star + \eta. \tag{22}$$

Here, $\eta$ is a 1-subgaussian random noise from the reward model, and $\phi : \mathcal{I} \times \mathcal{C} \to \mathbb{R}^d$ is the given feature map which sends text prompt $\mathbf{c} \in \mathcal{C}$ and generated image $\mathbf{x} \in \mathcal{I}$ into d-dimensional latent space.

**OLS estimator**   Suppose we have feature vectors $\phi_i(\mathbf{c}) := \phi(\mathbf{x}_i, \mathbf{c})$ and corresponding observations $r_i(\mathbf{x}, \mathbf{c})$ from Eq. 22 for $i = 1, 2, \ldots, n$. Then, we can estimate the true parameter $\theta_\star$ by following estimator which is known as OLS estimator:

$$\hat{\theta} = V^{-1} \sum_{i=1}^{n} r_i \phi(\mathbf{x}_i, \mathbf{c}), \tag{23}$$

where design matrix $V := \sum_{i=1}^{n} \phi_i(\mathbf{c})\phi_i(\mathbf{c})^\top$. For this $\hat{\theta}$, we can obtain following confidence bound for any $\phi_i \in \mathbb{R}^d$, $\delta \in (0, 1)$ as follows (Lattimore and Szepesvári (2020), Chapter 20):

$$\mathbb{P}\left( \langle \hat{\theta} - \theta_\star, \phi(\mathbf{x}, \mathbf{c}) \rangle \geq \sqrt{2\|\phi(\mathbf{x}, \mathbf{c})\|_{V^{-1}}^2 \log\left(\frac{1}{\delta}\right)} \right) \leq \delta. \tag{24}$$

Above equation implies, we can get better estimate $\hat{\theta}$ by minimizing $|\phi(\mathbf{x}, \mathbf{c})\|_{V^{-1}}$ and this problem is called as a G-optimal design:

**G-optimal design**   Let $\pi$ be a distribution on the collection of $\phi_i(\mathbf{c})$ and $\pi : \mathcal{A} \to [0, 1]$ be a distribution on $\pi$. The goal of G-optimal design is to find an optimal $\pi^\star$ which solves to find $\pi$ that minimizes the following objective:

$$g(\pi) = \max_{(i, \mathbf{c})} \|\phi_i(\mathbf{c})\|_{V(\pi)^{-1}}^2,$$

where, $V(\pi) = \sum_{i, \mathbf{c}} \pi(\phi_i)\phi_i(\mathbf{c})\phi_i(\mathbf{c})^\top$ is a design matrix constructed by the distribution $\pi$. Next, we introducte Kiefer-Wolfowitz theorem which is stated as follows.

**Theorem 2** (Kiefer-Wolfowitz). *Following statements are equivalent:*

  *(i)  $\pi^\star$ is a minimizer of $g$.*

  *(ii)  $\pi^\star$ is a maximizer of $f(\pi) = \log \det V(\pi)$*

  *(iii)  $g(\pi^\star) = d$.*

The proof of the theorem can be found in Lattimore and Szepesvári (2020). We can combine Theorem 2 with Eq. 24 to show that G-optimal design identifies an optimal data configuration that minimizes model prediction error under a limited budget. G-optimal design maximizes $\det V(\pi)$ by selecting $\phi_i$'s with diverse directions, implicitly incorporating diversity as an objective. Moreover, we can further explain the benefit of using high reward margin pairs by observing that while Eq. 24 tries to bound the all of the feature vectors uniformly, model model prediction error is most critical in the direction of $\theta_\star$. By collecting higher reward margin pairs, we can increase singular values of the design matrix $V(\pi)$ to the direction of $\theta_\star$. One can expect this will reduce the model prediction error for good feature vector, which in turn, results in better performance of the model. The further support of the claim can be linked to the literature of best arm identification (Yang and Tan, 2022).

## K   HUMAN EVALUATION

In this section, we provide detailed information about our human evaluation, briefly explained in Section 4.1. We randomly select 100 prompts for each concept in the HPSv2 benchmark, where the concepts are photo, paintings, anime, and concept art. For the selected 100 prompts, we use 50 prompts to create one survey form and the remaining 50 prompts for another, totaling 8 survey forms with 400 prompts. We assign three different annotators to each survey form with a total of 24 annotators.

In the case of human evaluations, we take extreme care to ensure that no harmful content is included on the survey form. All the authors cross-check the content to prevent any harmful material from being exposed to others. Moreover, we do not collect any personal information from participants, including their name, email, or any other identifying details. Instead, all results are gathered from anonymous users. To ensure this, we cross-check all options, remove any questions related to identity, and select options that do not collect any data (e.g., email) from the survey form. Additionally, among the eight test sets, each participant is assigned only one, making it unnecessary to collect private information such as an ID.

We have informed the participants about the purpose of our research, potential risks, and their right to withdraw their data. For the benefit of the participants, we provided a payment of 15\$ for each participant, where the evaluation took less than an hour. The instruction we provided to the participants is written as below:

---

**Instruction for Human Evaluation**

We are conducting a study to evaluate the performance of the advanced text-to-image diffusion models. In this survey, you will be presented with pairs of images that were generated based on specific captions. Your task is to compare each pair of images and select the one that you think best matches the caption or appears more visually appealing and coherent.

Before you begin, please be aware of the following important information:

Purpose of the Study: The aim of this study is to gather human feedback for the evaluation of certain text-to-image diffusion models.

Data Collection: We will collect data on your image selections you provide during the survey. This data will be used solely for research purposes of evaluating certain text-to-image generation models.

Potential Risk: While the study poses minimal risk, it is important to understand that your participation is not mandatory. Therefore, you have the right to withdraw from the study at any time, and you may request that your data be removed

By continuing with this survey, you acknowledge that you have read and understood the information provided above and consent to participate under these conditions. You have the right to ask any questions and receive clear answers before proceeding.

---

An example of the survey form is illustrated in Figure 13.

## L  MORE RESULTS

**Quantitative Impact of Text Diversity**  Diversity may play a more significant role when the original dataset contains a large number of duplicated text prompts. Therefore, we provide further quantitative results to demonstrate the importance of incorporating text diversity through a pilot experiment. Specifically, we select text prompts that are duplicated more than 20 times or share common keywords to construct a new dataset with multiple duplicated and similar prompts. We then apply `FiFA` with varying $\gamma$ values to analyze the impact of text diversity. As shown in Figure 14a, neglecting diversity reduces performance, as the model predominantly trains on similar prompts. In contrast, incorporating diversity by selecting a subset of distinct prompts from the original dataset achieves performance comparable to the full set, highlighting the importance of text diversity.

**Training without the Highest Margin Set**  To determine whether the highest margin set is necessary or if some level of high margin is sufficient, we compare the results of applying `FiFA` to the original dataset and a dataset excluding the top 10% margin pairs. As illustrated in Figure 14b, removing the top 10% significantly reduces performance, demonstrating that confident and clean pairs are essential for rapidly improving the model, particularly in the early stages of training, as they provide more informative signals.

**Results Using ImageReward**  To further demonstrate the generalizability of `FiFA`, we employ an alternative preference reward function, ImageReward (Xu et al., 2024), for calculating reward gaps and evaluation. As shown in Figure 14c, compared to random data selection or using the full dataset,

Please answer the following three questions, given two images **separated** by a **black gap**. You **must choose one** image that answers each question. If the decision is too close, please also select the **"tie"** option. **However, you still need to choose "left" or "right".** *

A moai wearing headphones.

|  | Left | Right | Tie (Please also choose "Left" or "Right") |
|---|---|---|---|
| Which image do you prefer given the caption? | ☐ | ☐ | ☐ |
| Which image is more visually appealing? | ☐ | ☐ | ☐ |
| Which image better fits the text description (caption)? | ☐ | ☐ | ☐ |

Figure 13: Example survey form for the human evaluation. We ask three questions about overall quality (general preference), image-only assessment, and text-image alignment. The images from `FiFA` and the full dataset are randomly assigned to the left or right. The annotators should choose either left or right, but they can also choose the tie option.

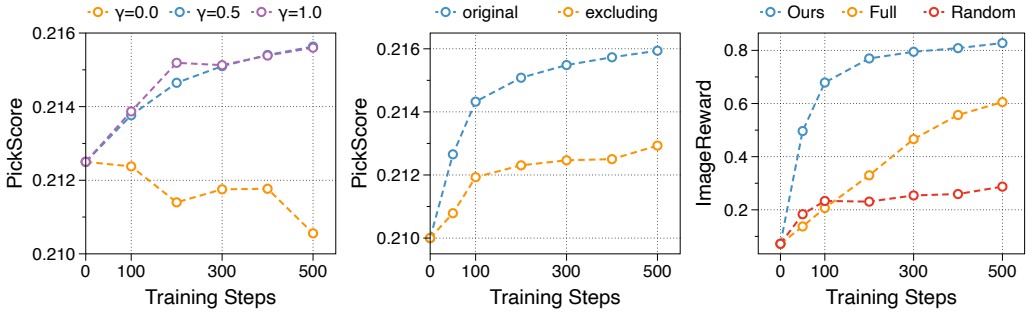

(a) Impact of Text Diversity  (b) Impact of the Highest Margin Set  (c) Results with ImageReward

Figure 14: (a) Impact of text diversity under specific settings. We created a subset of the Pick-a-Pic v2 dataset by selecting prompts with similar or identical keywords and compared **FiFA** across different levels of diversity by varying $\gamma$ (b) Results with and without filtering the top 10% margins of the training set before applying **FiFA**. (c) Comparison of random sampling (*R*), the full dataset (*full*), and **FiFA** with ImageReward (Xu et al., 2024) on the Pick-a-Pic v2 dataset.

**FiFA** rapidly improves the reward, demonstrating both its effectiveness and efficiency in terms of ImageReward values. These results indicate that our method is not reliant on a specific preference model.

## M  RELATED WORK

### M.1  ALIGNING TEXT-TO-IMAGE DIFFUSION MODELS

Fine-tuning text-to-image diffusion models using human feedback datasets has proven to be an effective way to improve large pretrained models. Early approaches are based on supervised fine-tuning, involving reward-weighted loss (Lee et al., 2023), or rejection-sampling based methods Dong et al. (2023); Sun et al. (2023). Policy gradient approaches are also proposed Black et al. (2023); Fan et al. (2024) to further improve the models with online learning. However, these methods are highly ineffective and severely limited to a small scale (Deng et al., 2024; Wallace et al., 2023). To address these issues, some methods directly use reward backpropagation (Clark et al., 2023; Prabhudesai et al., 2023), but these approaches are unstable and often lead to reward overoptimization (Zhang et al., 2024). Diffusion-DPO (Wallace et al., 2023), which directly utilizes human preferences, enables large-scale training without reward hacking, demonstrating its effectiveness for alignment along with its variants (Deng et al., 2024; Yang et al., 2023a). In this paper, we focus on Diffusion-DPO as it represents one of the most effective state-of-the-art methods for alignment using human feedback.

### M.2  DATA FILTERING FOR DEEP NEURAL NETWORKS

There have been attempts to improve the efficiency of training by selecting a core subset from the entire training data. Data pruning (Raju et al., 2021; Yang et al., 2022; He et al., 2023; Tan et al., 2024) involves removing unnecessary or less important data from a dataset to reduce its size and enhance efficiency. Coreset selection (Xia et al., 2022; Mirzasoleiman et al., 2020; Gupta et al., 2023; Chai et al., 2023) aims to select a small subset that retains the original dataset's statistical properties. Data distillation (Cazenavette et al., 2022; Nguyen et al., 2020; Bohdal et al., 2020) and data condensation (Liu et al., 2023; Kim et al., 2022; Wang et al., 2022) generate small synthetic training data from the entire dataset that can achieve the learning effect of the entire data. However, these approaches usually estimate uncertainty based on gradients or loss, often do not align well with diffusion models, as calculating the loss or gradient for diffusion models is time-consuming. In the LLM domain, Zhou et al. (2024) introduced a small yet effective dataset for instruction tuning, while Ko et al. (2025), similar to our work, selected samples with large implicit reward margins during DPO training. In diffusion domains, Emu (Dai et al., 2023) demonstrated that training on a high-quality, curated dataset significantly improves image quality and text-image alignment. However, its reliance on extensive human curation makes it impractical for large-scale applications. Unlike these approaches, we propose an automated data filtering framework that minimizes human involvement.

# N    MORE EXAMPLES

Figure 15-18 demonstrate more randomly selected examples generated using the Pick-a-Pic test set, PartiPrompt, and HPSv2 benchmark.

| **Pretrain** | **DPO + Full** | **DPO + FiFA** |
|---|---|---|

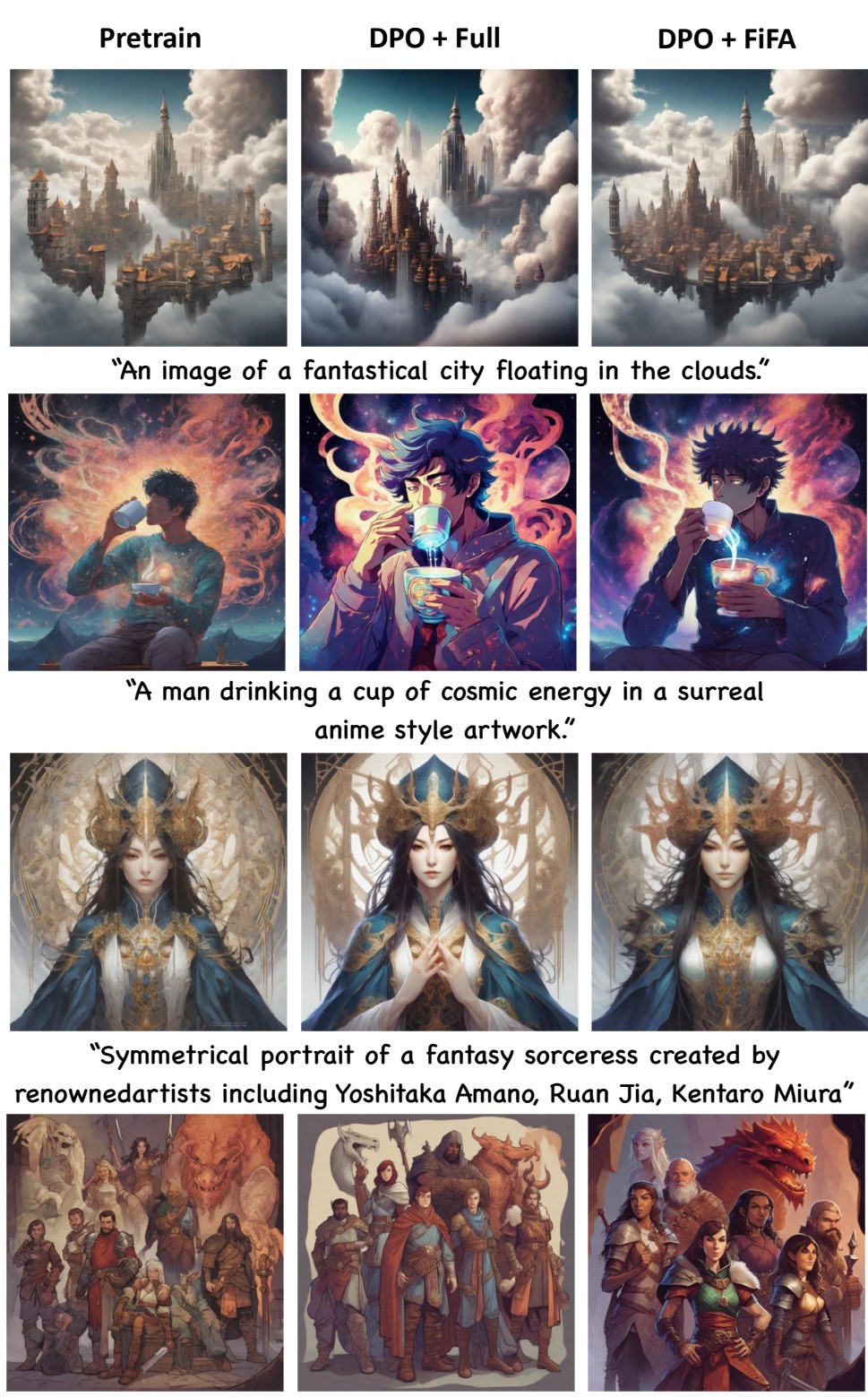

"An image of a fantastical city floating in the clouds."

"A man drinking a cup of cosmic energy in a surreal anime style artwork."

"Symmetrical portrait of a fantasy sorceress created by renownedartists including Yoshitaka Amano, Ruan Jia, Kentaro Miura"

"An illustration featuring characters from a Dungeons and Dragons game."

Figure 15: More example images generated using the different models.

| Pretrain | DPO + Full | DPO + FiFA |
|----------|-----------|-----------|

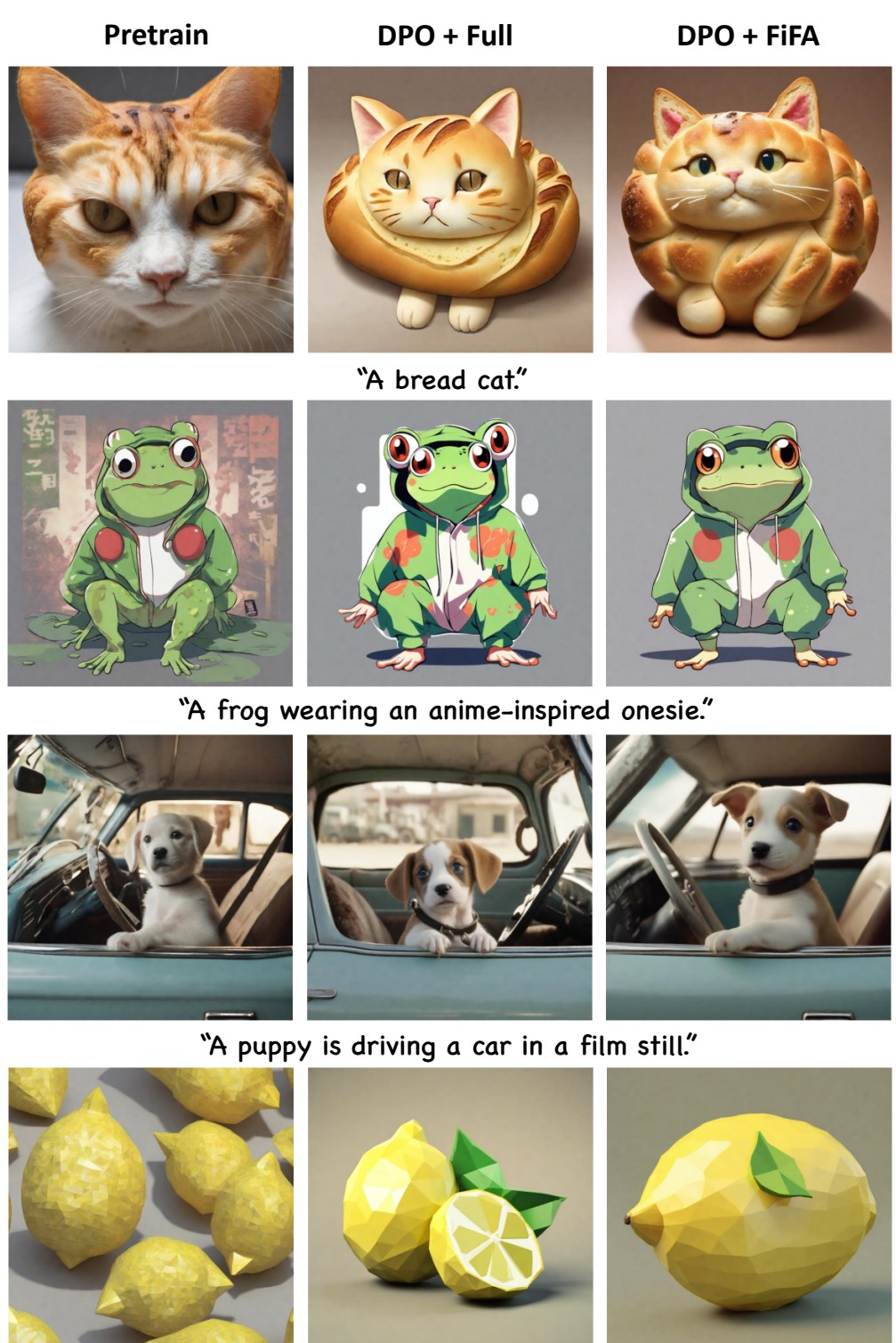

Figure 16: More example images generated using the different models.

**Pretrain**       **DPO + Full**       **DPO + FiFA**

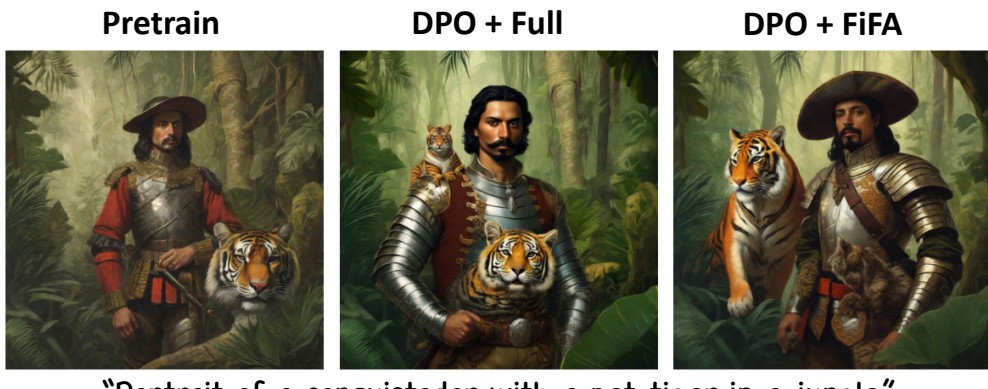

"Portrait of a conquistador with a pet tiger in a jungle."

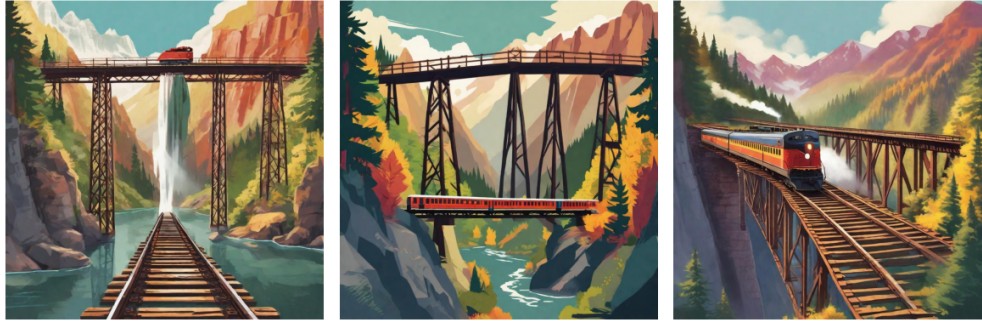

"A train crosses a trestle bridge in the mountains in
an optimistic and vibrant illustration."

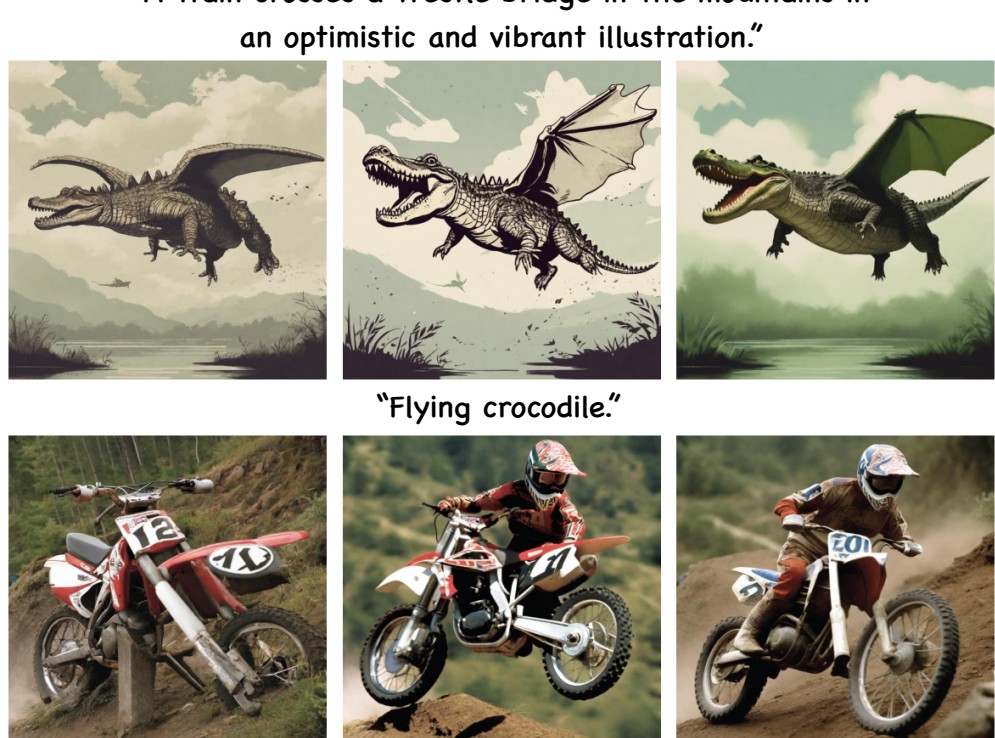

"Flying crocodile."

"The dirt bike has seen many hill climbs in its history."

Figure 17: More example images generated using the different models.

| Pretrain | DPO + Full | DPO + FiFA |
|----------|------------|------------|

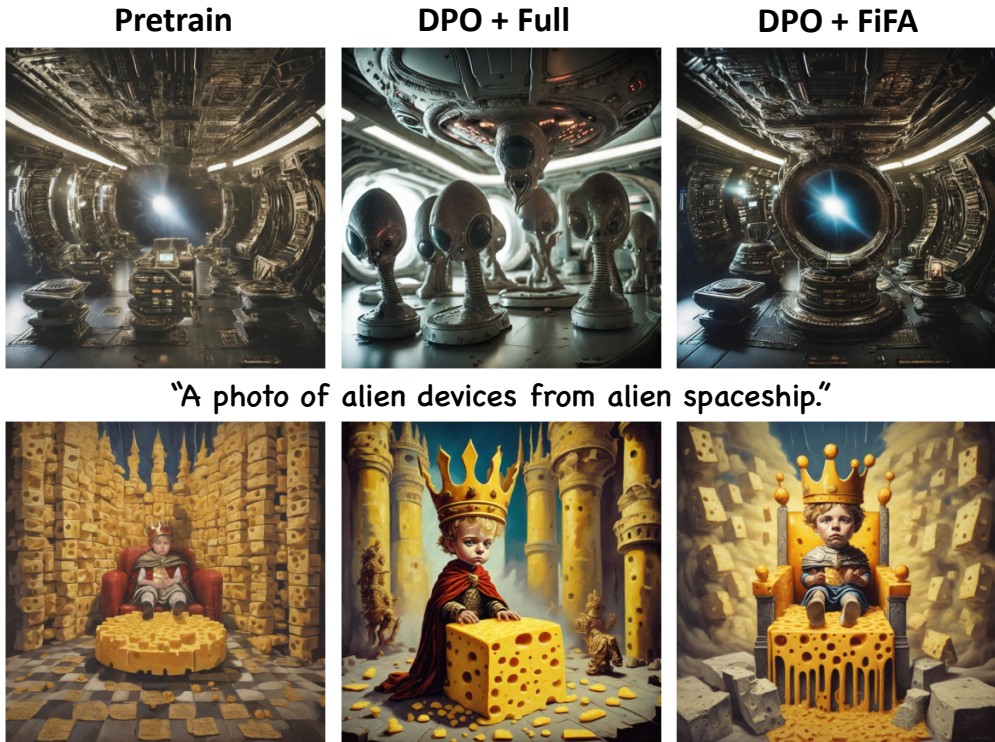

"A photo of alien devices from alien spaceship."

"An oil painting of a child king ruling a kingdom made entirely of cheese in a surreal and comic book style."

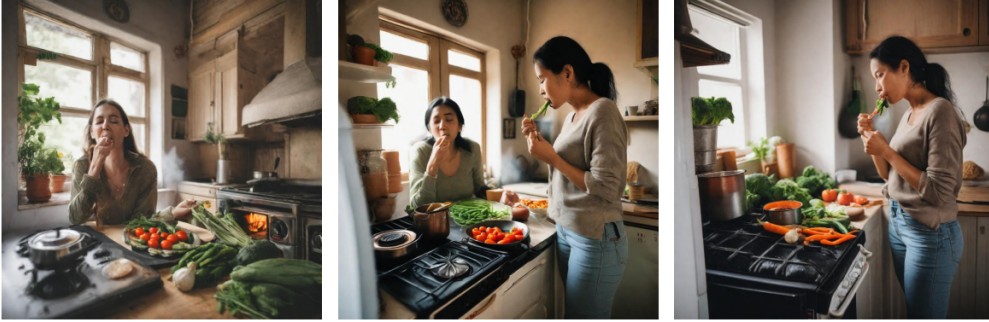

"A woman eating vegetables in front of a stove."

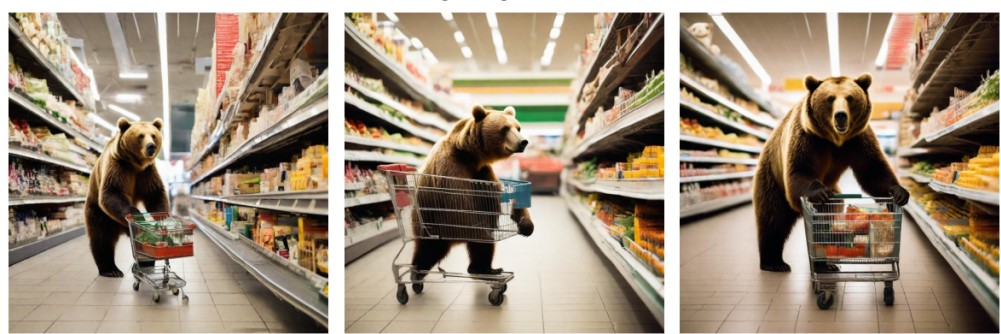

"A brown bear pushes a shopping cart in a grocery store."

Figure 18: More example images generated using the different models.

