# OpenReview forum: "Automated Filtering of Human Feedback Data for Aligning Text-to-Image Diffusion Models"
_ICLR.cc/2025/Conference — ICLR 2025 Poster_

### Official Review · Reviewer_CrC4 · 2024-10-15

**Soundness:** 3
**Presentation:** 3
**Contribution:** 3
**Rating:** 6
**Confidence:** 3

**Summary:**

This paper proposes FiFA (Filtering for Feedback Alignment), a novel automated data filtering approach for efficiently fine-tuning text-to-image diffusion models using human feedback data.

The main contributions are:
* An automated data selection method that maximizes: preference margin, text quality and text diversity.
* Formulation of data selection as an optimization problem to find a subset that maximizes these components.
* Empirical evidence showing FiFA's effectiveness across various models and datasets.

**Strengths:**

The paper presents a novel approach to data filtering for fine-tuning text-to-image diffusion models. While data pruning and coreset selection are not new concepts in the domain of text-to-image diffusion models (first documented by Meta’s EMU paper), this work focuses on the automation of coreset selection. The combination of preference margin, text quality, and text diversity in a single optimization framework is an effective and reasonable solution in this problem space.

The paper demonstrates effective results across different models (SD1.5 and SDXL) and datasets (Pick-a-Pic v2 and HPSv2), providing robust evidence for their claims. The inclusion of both automatic metrics and human eval provide a complete picture in terms of metrics. There is also some  theoretical analysis provided in the author’s paper.

its most impressive for the authors to achieve high quality alignment with just 0.5% of the data and 1% of the GPU hours. FiFA also demonstrated reduction in harmful content generation, which is critical for these automatic coreset selection method.

**Weaknesses:**

I think the biggest issue with this work is that it did not experiment with strong diffusion models like SD3-2B or FLUX models or the Playground models. Those models are much better to start with. It would be very helpful to know if the proposed model can further improve strong models.

**Questions:**

The authors highlighted that the method can achieve good results with just 0.5% of the data. Do you have results showing how well FiFA filtering works on say 0.1%, 1%, 5%, 10% of the dataset? It could help us understand how tunable FiFA is.

---

> ### Author Response · Authors · 2024-11-20
> **Response to Reviewer CrC4**
>
> Thank you so much for recognizing the effectiveness of our approach and providing valuable feedback. We have simplified your comments for clarity and provided our corresponding responses. We look forward to continuing our discussion.
>
> ---
>
> ### ***W1: Lack of Experiments on Experiment with Strong Diffusion Models***
>
> Thank you so much for your suggestion. We agree that it would be great to see the results on the latest models, such as SD3 or FLUX, based on diffusion transformers. To address your concern, we conducted experiments on the SD3-2B model using the Pick-a-Pic v2 dataset for training. The FiFA model is trained for 1 epoch, whereas the full dataset model is trained for only 0.1 epoch due to time constraints (equivalent to training on a random 10% subset of the full dataset). Consequently, the FiFA model utilizes approximately 10% of the GPU hours compared to training with the full dataset. The results are shown below:
>
> | Method       | PickScore | Win Rate over Pretrain |
> |--------------|-----------|------------------------|
> | Pretrain     | 22.076       |         N/A           |
> | DPO + full   | 22.085       | 54.1 %                    |
> | DPO + FiFA   | **22.213**       |         **59.2 %**               |
>
> As shown in the results, our method remains effective on the SD3-2B model, demonstrating its generalizability across different architectures. Additionally, we believe the results could be **significantly improved with a new preference dataset that includes higher-quality images**, such as those from FLUX or SD3, since the highest image quality in Pick-a-Pic v2 comes from SDXL. But still, it is surprising that our method performs well in this case, likely due to the training nature of DPO.
>
> ---
>
>
> ### ***W2: Experiments on Different Number of Filtered Dataset using FiFA (e.g. 0.1%, 0.5%, …)***
>
> We completely agree that it is important to analyze the results with different amounts of filtered data and we have **already conducted an ablation study, as illustrated in ${\color{blue}\text{Figure 6(c)}}$ of our main paper**. In our main experiments, we set the number of data points K to 5000, which K=1000 represents 0.1%, and K=50000 represents 5%.
>
> As shown in this figure and described in our paper, there are two key observations:
> ***1)*** Across the tested range (0.1% to 5%), all results exceed the performance of using the full dataset, indicating that the method does not overly depend on a specific K value. ***2)*** Including more data is not always beneficial, as it introduces more noise. Conversely, using too little data may also be detrimental due to insufficient information.
>
> However, the trend might vary depending on the original dataset. For instance, if the original dataset consists entirely of high-quality data, including more data might be helpful.

---

> ### Author Response · Authors · 2024-11-25
> **Kind reminder to Reviewer CrC4**
>
> Dear Reviewer CrC4,
>
> Thank you again for your time and efforts in reviewing our paper.
>
> As the discussion period draws close, we kindly remind you that two days remain for further comments or questions. We would appreciate the opportunity to address any additional concerns you may have before the discussion phase ends.
>
> Thank you very much!
>
> Many thanks, Authors

---

> ### Author Response · Authors · 2024-11-27
> **Gentle Reminder to Reviewer CrC4**
>
> Dear Reviewer **CrC4**,
>
> Thank you once again for your time and effort in reviewing our paper. We greatly appreciate your valuable feedback and suggestions.
>
> We would like to gently remind you that the discussion period is coming to a close.
>
> In our rebuttal, we have:
>
> - **Demonstrated the results using SD3-2B.**
> - **Clarified the ablation study on the percentage of selected data.**
>
>
> If you have any remaining concerns, please do not hesitate to share them with us. We are more than willing to address them promptly.
>
> Thank you very much for your consideration.
>
> Best regards,
>
> Authors

---

> ### Author Response · Authors · 2024-11-29
> **Gentle Reminder to Reviewer CrC4**
>
> Dear Reviewer CrC4,
>
> We truly appreciate your time and effort in reviewing our work.
>
> As the discussion period is nearing its end, we kindly remind you that only a few days remain for further comments or questions.
>
> In response to your feedback, we have provided a detailed response and added a summary of our response recently.
>
> We kindly ask if you have any additional concerns or questions that we may address during the remaining discussion period.
>
> Thank you once again for your valuable insights.
>
> Best regards, \
> The Authors

---

> ### Author Response · Authors · 2024-12-02
> **Final Kind Reminder: Review Discussion Period – One Day Remaining**
>
> Dear Reviewer CrC4,
>
> We wanted to kindly remind you that only one day remains in the review discussion period.
>
> We hope that our response and revised manuscript have provided the necessary information to address your questions. If you have had a chance to review our response, we would greatly appreciate it if you could confirm this and let us know if there are any additional questions or concerns that we can address before the discussion period concludes.
>
> If there are no further questions, we hope our revisions and responses have satisfactorily addressed your feedback and would be grateful if this could be reflected in your re-evaluation.
>
> Thank you once again for your time and consideration.
>
> Best regards,
> The Authors

---

> > ### Comment · Reviewer_CrC4 · 2024-12-02
> >
> > Thank you for the response.

---

> > > ### Author Response · Authors · 2024-12-02
> > > **Thank you for your response – Follow-Up on Concerns and Re-Evaluation for Reviewer CrC4**
> > >
> > > Dear Reviewer CrC4,
> > >
> > > Thank you so much for your response. We hope that our previous reply has adequately addressed all your concerns. If there are any additional questions or points you'd like us to clarify, please let us know before the discussion period ends.
> > >
> > > If all your concerns have been resolved, we would greatly appreciate it if you might consider re-evaluating your review.
> > >
> > > Sincerely,
> > > The Authors

---

### Official Review · Reviewer_irH1 · 2024-10-30

**Soundness:** 2
**Presentation:** 3
**Contribution:** 2
**Rating:** 6
**Confidence:** 5

**Summary:**

In this paper, the authors aim to improve aligning text-to-image diffusion models from the perspective of filtering human feedback data. Specifically, they select the data pairs by maximizing three components: preference margin, text quality, and text diversity. For each component, they design an optimization objective. Finally, several experiments have been conducted to verify the contribution of each component

**Strengths:**

1.	The motivation of filtering the human feedback data is reasonable. It is well-known that the training of diffusion is very cost. High quality data would contribution both the effectiveness and efficiency of the model.

2.	The paper writing is great.

**Weaknesses:**

1.	The technical contribution is relatively small. In my opinion, the proposed approaches for filtering data are travel and nature. In addition, as for preference margin, I believe that it is better to maximize preference margin in a limited range, while a very large margin would provide difficulty for optimization.

2.	Only the pick-a-pic dataset is used in the experiment, which is highly related to Pick Score. Some other datasets should be involved to verify the generalization. For example, the authors can use the HPS score to compute the preference margin, even on the same pick-a-pic dataset.

3.	It is also significant to show the pair-wise human evaluation in Figure 4.

**Questions:**

How about the effectiveness of the proposed approaches on other datasets with different preference models?

I also want to the win rate of the proposed approaches compared with only the base model or DPO.

---

> ### Author Response · Authors · 2024-11-20
> **Response to Reviewer irH1**
>
> We greatly appreciate your thoughtful feedback and critical advice to improve our paper. We have simplified your comments for easier reference and included our respective responses. We will carefully address your concerns one by one.
>
> ---
>
> ### ***W1: Technical Contribution and Concerns on High Preference Margin***
>
> To address your concerns, we would like to clarify and emphasize our contributions:
>
> ***Filtering Criteria***: Our method filters data based on three criteria: reward gap, text quality, and diversity. To our knowledge, this is the first application showing high performance in Diffusion-DPO through selection at high reward gap points. While reward gap is a significant factor, we also integrate text quality and diversity to mitigate risks and enhance generalizability. The combination of diversity and reward gap is also grounded in the principles of the G-optimal design in theory of experimental design.
>
> ***Optimization Strategy***: Setting fixed thresholds for each filtering criterion can result in data scarcity or over-reliance on the original dataset. To counter this, we formulated an optimization problem with a practical approximation, enabling us to efficiently select *K* optimal data points, balancing all three factors.
>
> ***High Preference Margins and Optimization***: Contrary to concerns about high preference margins complicating optimization, our approach leverages the DPO framework’s reliance on deterministic preference training rather than direct reward values. Using data with larger reward gaps actually supports more stable dynamics early in training, enhancing efficiency.
>
> To demonstrate this, we tested our approach by excluding the top 10% of data points with excessively high preference margins before applying the FiFA algorithm, comparing it with the original method including these points:
>
> | Method                    | 100      | 300      | 500      |
> |---------------------------|----------|----------|----------|
> | Excluding Top 10% Margin    | 21.192   | 21.247   | 21.293   |
> | Including Top 10% Margin (Original FiFA)    | 21.432   | 21.548   | **21.594**   |
>
> Detailed results are presented in ${\color{blue}\text{Figure 14 (b)}}$ of ${\color{blue}\text{Appendix L}}$. These results demonstrate that limiting reward gaps does not enhance performance and may reduce it, supporting our approach of not restricting high reward gaps, which we found does not pose optimization challenges, particularly in the early stages of training.
>
> ---
>
> ### ***W2: Lack of Experiments using Other Preference Models, such as HPS***
>
>
> We would like to clarify that **we reported the result of experiments using HPS v2 reward**. Our paper proposes that training a reward model on the full dataset, filtering the dataset with this reward model and other components, and then fine-tuning using DPO is an efficient approach. Accordingly, we used **PickScore for training on the Pick-a-Pic v2 dataset and HPSv2 reward for training on the HPS v2 trainset** in the main experiment shown in ${\color{blue}\text{Table 1}}$. We added this description in the revised manuscript.
>
> In addition, we report the results of using another preference model, **ImageReward [1]**, instead of PickScore, for training on the Pick-a-Pic v2 dataset. The model was trained for 500 steps using the SD1.5 model. The results are as follows:
>
> | Method       | ImageReward Score |
> |--------------|--------------------|
> | Pretrain     | 0.07              |
> | DPO + Full   | 0.61              |
> | DPO + FiFA   | **0.83**              |
>
> Detailed results are presented in ${\color{blue}\text{Figure 14 (c)}}$ of ${\color{blue}\text{Appendix L}}$. Overally, FiFA outperforms the full dataset across all three preference models, demonstrating that its effectiveness does not depend on a specific reward model.
>
>
> ### References
> [1] Xu et al. “ImageReward: Learning and Evaluating Human Preferences for Text-to-Image Generation,” NeurIPS 2023.
>
> ---
>
>
>
> ### ***W3:  About Human Evaluation.***
>
>
> Thank you for your questions regarding our human evaluation and pair-wise comparison. We will first clarify the process of our human evaluation. As shown in ${\color{blue}\text{Figure 13}}$ of ${\color{blue}\text{Appendix K}}$, we presented a pair of images, both generated using the DPO method—one with our filtered data and one with the full data. Users were asked to choose their preferred image or select a tie option. The win rate is therefore a **pair-wise evaluation** over the DPO method (*i.e.*  Base DPO) using the full dataset as a baseline.
>
> If we have misunderstood the meaning of “pair-wise evaluation” or your feedback, please let us know—we are open to further discussion!

---

> ### Author Response · Authors · 2024-11-25
> **Kind reminder to irH1**
>
> Dear Reviewer irH1,
>
> Thank you again for your time and efforts in reviewing our paper.
>
> As the discussion period draws close, we kindly remind you that two days remain for further comments or questions. We would appreciate the opportunity to address any additional concerns you may have before the discussion phase ends.
>
> Thank you very much!
>
> Many thanks, Authors

---

> > ### Comment · Reviewer_irH1 · 2024-11-26
> >
> > The responses address most of my concerns. I have raised my score.

---

> > > ### Author Response · Authors · 2024-11-26
> > > **Thanks for the Feedback and Support**
> > >
> > > Dear Reviewer irH1,
> > >
> > > We are so delighted to hear that most of your concerns have been addressed. Thank you once again for the time and effort you have dedicated to reviewing our paper.
> > >
> > > Best regards,
> > > The Authors

---

### Official Review · Reviewer_f1jQ · 2024-11-03

**Soundness:** 3
**Presentation:** 3
**Contribution:** 2
**Rating:** 6
**Confidence:** 3

**Summary:**

This work presents a novel approach to fine-tuning text-to-image diffusion models using human feedback data filtered through an automated algorithm. The proposed methodology optimizes the fine-tuning process by selecting a subset of the available human feedback based on a preference margin criterion, enhancing the reward value while considering both prompt quality and diversity to maintain robustness and mitigate potential harmful content.

**Strengths:**

Please see questions

**Weaknesses:**

Please see questions

**Questions:**

1. The proposed filtering algorithm systematically narrows down the human feedback dataset to a subset that is optimal for model fine-tuning. As a general approach, further discussion on the generalizability of this filtering approach could enrich the analysis, such as how it may integrate with other alignment frameworks like RLHF and DPO-based methods. In addition, expanding the range of comparative methods would strengthen the evaluation.

2. To clarify the novelty of this approach, the specific roles of preference margin and the quality/diversity metrics for text prompts could be further justified. Detailing the design motivations behind these components and their interdependencies would clarify their contributions to the model’s overall performance.

3. DPO requires extensive high-quality preference data, which can be costly and difficult to obtain. The accuracy of preference data is essential, as low-quality feedback may lead to biased or suboptimal model behavior. There appears to be some ambiguity in the statement regarding dataset preparation: "To ensure safety, we manually filter out some harmful text prompts from these test prompts, resulting in 446 unique prompts." It seems that an additional manual filtering step was applied before evaluating the proposed algorithm’s ability to handle harmful prompts. Clarifying this step’s rationale and how it affects the filtering method’s efficacy would add clarity.

4. DPO-based approaches can sometimes narrow the scope of outputs, potentially limiting diversity. To validate the claimed advantage of this filtering method in maintaining diverse outputs, more empirical evidence should be presented. Additionally, a comparison with online DPO and recent DPO variants would help contextualize the findings.

5. More qualitative evidence on how the proposed approach reduces training costs would be valuable, particularly with concrete examples or case studies showing the efficiency gains obtained through this filtering algorithm.

6. Human evaluation is conducted in this work, however, it appears that there is no evidence of human ethics approval, despite it potentially being a low-risk case.

**Details Of Ethics Concerns:**

Human evaluation is conducted in this work, however, it appears that there is no evidence of human ethics approval, despite it potentially being a low-risk case.

---

> ### Author Response · Authors · 2024-11-20
> **Response to Reviewer f1jQ (1/2)**
>
> Thank you so much for providing thoughtful and helpful feedback. We have simplified your comments for easier reference and included our respective responses. We will carefully address your concerns one by one.
>
> ---
>
> ### ***W1 : Effectiveness on Other RLHF or More Results on Online DPO or DPO Variants***
>
> Thank you for coming up with such exciting discussion points. We think that our filtering approach, FiFA, can be effectively combined with other model alignment methods, such as online DPO, its variants, or other RLHF methods. Specifically, for DPO variants that utilize preference datasets, FiFA integrates seamlessly. For online DPO, FiFA can be applied iteratively after generating online samples with current models to continuously refine the data.
>
> Additionally, we suggest applying FiFA to PPO by strategically selecting text prompts for training. This can be achieved by measuring the margins of online samples and evaluating LLM scores. We believe that such an extension would not only be intriguing but also significantly enhance the value of our FiFA framework.
>
> ---
>
> ### **W2 : Specific Roles of Preference Margin, Text Quality, and Text Diversity**
>
>
> In the **Introduction (Section 1)** (lines 89–92) and **Method (Section 3)**, we explain the role of each component. Below, we summarize each role:
>
> - ***Preference Margin***: This is the primary component for efficiently and effectively increasing the reward. Due to the noisy nature of preference datasets, having a clear margin significantly improves performance. The independent effect of the preference margin is shown in ${\color{blue}\text{Figure 6(b)}}$
>
> - ***Text Quality***: Text quality is crucial for improving safety, as the naive use of open-source data can lead to serious issues, such as NSFW content. Additionally, text quality slightly enhances reward performance by removing meaningless prompts. The independent effect of text quality is shown in ${\color{blue}\text{Figure 7(b)}}$.
>
> - ***Text Diversity***: Text diversity aids in generalization and improves performance on diverse prompts by ensuring sufficient coverage. The independent effect of text diversity is shown in ${\color{blue}\text{Figure 7(c)}}$. To further highlight the impact of diversity, we also report the win rate of FiFA over DPO with and without considering text diversity on the PartiPrompt dataset. This dataset belongs to a different domain from the training set of Pick-a-Pic v2, and the results are as follows:
>
> | Method           | Win Rate over base DPO |
> |-------------------|----------|
> | Without Diversity | 68.2%   |
> | With Diversity    | **71.7%**   |
>
> If you need further clarification, we are happy to discuss this in more detail.
>
> ---
>
> ### ***W3 : Concerns on Filtering the Pick-a-Pic test set.***
>
>
> We clearly understand your concern regarding the ambiguity. In the main tables (*e.g.* ${\color{blue}\text{Table 1}}$), to primarily evaluate text-image alignment and image quality, we filter out a small number (54, 10%) of highly harmful prompts (*e.g.* “Nakxx girl with tixx”) that could lead to potential harm or additional safety issues, since we aim to separate experiments for text-image alignment and aesthetics from safety experiments. Therefore, we conduct dedicated experiments to evaluate how models handle safety issues, presenting the results in ${\color{blue}\text{Figure 7(b)}}$ under controlled conditions to avoid exposing these harms to others.
>
>
> Furthermore, we applied filtering only to the Pick-a-Pic v2 test set, as the PartiPrompt and HPS v2 benchmarks do not exhibit the same issues. Despite this, FiFA performs well across all three benchmarks.

---

> ### Author Response · Authors · 2024-11-20
> **Response to Reviewer f1jQ (2/2)**
>
> ### ***W4 : Output (Image) Diversity with FiFA***
>
>
> To address your concern, we assessed pair-wise image diversity using the CLIP ViT encoder. Specifically, we generated multiple images using different seeds for each prompt, calculated the embeddings, created a similarity matrix for each prompt, and computed the average similarity. These values were then averaged across all prompts. The resulting similarity scores are as follows:
>
> | Method   | Avg. Distance (Diversity) |
> |----------|--------------------|
> | Pretrain | 0.262      |
> | Full     | 0.251      |
> | Ours     | 0.235      |
>
> The output diversity is largely maintained, although there is a slight reduction in diversity as we aim to align the model output more closely with human preferences.
>
> ---
>
>
> ### ***W5 : Qualitative Evidence on Efficiency***
>
> To address your concern, we will specifically analyze the GPU hours required by our framework and compare them with the time needed to train the model using the full dataset. We use SDXL models, and all GPU hours refer to usage on A6000 GPUs.
>
> For the Pick-a-Pic v2 dataset, the reward model training using CLIP architecture takes 4.3 hours (this step could be skipped by using open-source reward models), filtering requires less than 1 hour (including calculating rewards for each image), and training for 500 steps (~10 epochs) takes 13 hours, totaling approximately 18.3 hours. In comparison, training on the full dataset for just 1 epoch requires 656 hours, while completing 1,000 steps using the Hugging Face checkpoint of Diffusion-DPO takes 1,760 hours, which far exceeds the total time required using FiFA. This demonstrates that reducing the dataset size significantly improves efficiency.
>
>
> If this does not provide the qualitative evidence you are seeking, please let us know, as we are open to further discussion.
>
> ---
>
>
> ### ***W6 : Ethical Issue***
>
> We understand your concern regarding the need for careful handling of human evaluations. We did not get approval as our study involves human raters anonymously evaluating image preferences from a benchmark dataset and there are no potential risks to the participants in this study. Moreover, as demonstrated in **Appendix K**, we have included detailed instructions for users, which clearly outline the purpose of the study, the preservation of privacy, and their right to refuse participation or delete their annotations. Additionally, all authors have cross-checked for any potential harm or privacy issues related to the evaluation.

---

> ### Author Response · Authors · 2024-11-25
> **Kind reminder to Reviewer f1jQ**
>
> Dear Reviewer f1jQ,
>
> Thank you again for your time and efforts in reviewing our paper.
>
> As the discussion period draws close, we kindly remind you that two days remain for further comments or questions. We would appreciate the opportunity to address any additional concerns you may have before the discussion phase ends.
>
> Thank you very much!
>
> Many thanks,
> Authors

---

> ### Author Response · Authors · 2024-11-27
> **Gentle Reminder to Reviewer f1jQ**
>
> Dear Reviewer **f1jQ**,
>
> Thank you once again for your time and effort in reviewing our paper. We greatly appreciate your valuable feedback and suggestions.
>
> We would like to gently remind you that the discussion period is coming to a close.
>
> In our rebuttal, we have:
>
> - **Incorporated discussions on the online DPO, DPO variants, and other RLHF in the revised PDF.**
> - **Clarified the novelty and roles of each component of FiFA.**
> - **Clarified the rationale behind test filtering.**
> - **Added results on image diversity.**
> - **Provided qualitative evidence on efficiency.**
> - **Justified human evaluation.**
>
> If you have any remaining concerns, please do not hesitate to share them with us. We are more than willing to address them promptly.
>
> Thank you very much for your consideration.
>
> Best regards,
>
> Authors

---

> ### Author Response · Authors · 2024-11-29
> **Gentle Reminder to Reviewer f1jQ**
>
> Dear Reviewer f1jQ,
>
> We truly appreciate your time and effort in reviewing our work.
>
> As the discussion period is nearing its end, we kindly remind you that only a few days remain for further comments or questions.
>
> In response to your feedback, we have provided a detailed response and added a summary of our response recently.
>
> We kindly ask if you have any additional concerns or questions that we may address during the remaining discussion period.
>
> Thank you once again for your valuable insights.
>
> Best regards, \
> The Authors

---

> ### Author Response · Authors · 2024-12-02
> **Final Kind Reminder: Review Discussion Period – One Day Remaining**
>
> Dear Reviewer f1jQ,
>
> We wanted to kindly remind you that only **one day remains** in the review discussion period.
>
> We hope that our response and revised manuscript have provided the necessary information to address your questions. If you have had a chance to review our response, we would greatly appreciate it if you could confirm this and let us know if there are any additional questions or concerns that we can address before the discussion period concludes.
>
> If there are no further questions, we hope our revisions and responses have satisfactorily addressed your feedback and would be grateful if this could be reflected in your re-evaluation.
>
> Thank you once again for your time and consideration.
>
> Best regards, \
> The Authors

---

### Official Review · Reviewer_uZ1n · 2024-11-04

**Soundness:** 3
**Presentation:** 3
**Contribution:** 2
**Rating:** 5
**Confidence:** 4

**Summary:**

This paper introduces FiFA, an automated data filtering algorithm designed to optimize fine-tuning of text-to-image diffusion models, aligning model behavior more effectively with human intent. While human feedback datasets are valuable for model alignment, their large size and high noise levels often hinder convergence. FiFA enhances fine-tuning by automatically filtering data based on an optimization problem that maximizes three key components: preference margin, text quality, and text diversity. Experimental results show that FiFA enhances training speed and achieves better performance.

**Strengths:**

1. The paper propose the FiFA algorithm, which leverages three core metrics—preference margin, text quality, and text diversity—to optimize data filtering automatically. This approach effectively addresses noise in human feedback datasets and improves the fine-tuning of diffusion models, particularly for large-scale datasets.
2. The paper is of high quality, with well-designed and comprehensive experiments, including several ablation and comparative studies that strongly support the effectiveness of FiFA in enhancing training efficiency and image quality.
3. The structure of the paper is clear and well-organized. Key concepts, such as preference margin, text quality, and text diversity, are clearly defined, making the methodology accessible.

**Weaknesses:**

1. In the paper, some equations lack corresponding equation numbers.
2. In the introduction, the phrasing around "difficulty of convergence" is inconsistent with the discussion of the iterative training required for diffusion models. It is recommended that the authors clarify the logical flow.
3. In Equation 2, there is an extra left parenthesis "(".
4. When using pre-trained models (e.g., CLIP, BLIP) to calculate the preference margin, how is the validity of the results ensured? Given that pre-trained models are trained on noisy and ambiguous datasets, they may also yield incorrect results.
5. From the ablation study results, the effects of Text quality and Text diversity are not very significant. The authors state, "when combined with a high margin, they outperform the model trained solely on margin, highlighting the importance of both components," but where are the results? Is the improvement due solely to the higher margin?

**Questions:**

Please refer to Weaknesses.

---

> ### Author Response · Authors · 2024-11-20
> **Response to Reviewer uZ1n (1/2)**
>
> Thank you so much for clearly understanding our paper with strengths recognized and provided helpful feedback. We will address your concerns one by one. We have simplified your comments for easier reference and included our respective responses.
>
>
> ---
>
> ### ***W1:  About Equation Numbers and Parenthesis***
>
> Thank you so much for pointing out the issues with some equations. We have identified all the relevant parts and fixed these parts in the revised version.
>
> ---
>
> ### ***W2:  Inconsistency of “difficulty in convergence” and “iterative training of diffusion models.”***
>
> Thank you for discussing these points. To address your concern, we want to clarify that these **two concepts are not related to the inconsistency** of logical flow. Assuming that "iterative training" refers to line 53, where we mention "must be trained on multiple timesteps," we provide more details on the two arguments: “difficulty in convergence” and “iterative training of diffusion models.”
>
>
> During the diffusion training process, timesteps are uniformly sampled. To train the model sufficiently, the same data point should be trained at different timesteps by increasing the number of epochs. This does not imply that the model must be trained iteratively in a sequential manner. Additionally, the difficulty in convergence refers to the challenge of achieving convergence during training on multiple timesteps, particularly when preferences are noisy. This challenge is illustrated in ${\color{blue}\text{Figures 11(a)}}$ and ${\color{blue}\text{11(b)}}$ in ${\color{blue}\text{Appendix G}}$, where the model struggles to reduce the loss and improve implicit reward accuracy. Therefore, these **two concepts can coexist**, and one does not necessarily preclude the other.
>
>
> If our assumption is incorrect, if we misunderstood your feedback, or if this explanation is unclear, please feel free to ask additional questions. We are happy to engage in further discussion.
>
>
>
> ---
>
> ### ***W3: Problems with Pretrained CLIP and BLIP Models for Calculating Margin***
>
>
> Thank you for raising these important points. However, **we want to clarify that we do not use the pretrained CLIP or BLIP models directly**. Instead, we use fine-tuned reward models trained on human feedback datasets to calculate the preference margin, leveraging only the architectures of CLIP and BLIP.
>
>
> To be more specific, our filtering process involves using reward models trained on the entire dataset with reward training or open-source reward models. Therefore, we do not rely on pretrained CLIP or BLIP but rather on reward-trained models, such as PickScore and HPSv2 model.

---

> ### Author Response · Authors · 2024-11-20
> **Response to Reviewer uZ1n (2/2)**
>
> ### ***W4: Effects of Text Quality and Diversity***
>
> Thank you for bringing up the important points. Although preference margin is primarily used for increasing the reward, as mentioned in the Introduction (Section 1, lines 92-94) and detailed in the Method section (Section 3), **the main purpose of considering text quality and diversity is not just increasing the reward**. Specifically, considering text quality significantly **reduces harmfulness**, while diversity **enhances generalization capability** (increase rewards on more diverse concepts). The full results and detailed explanations of each component’s contribution are provided in ${\color{blue}\text{Figure 7}}$ and ${\color{blue}\text{Section 4.4}}$. To rapidly increase the reward value on the Pick-a-Pic v2 dataset, the margin plays the most critical role. Since the original sentence, “highlighting the importance of both components," could be misleading, we revised it to: “suggesting that sacrificing some margin for higher text diversity and quality could slightly boost performance while providing additional benefits."
>
>
>
> Additionally, as the results on diversity show qualitative improvement, we also report the win rate of FiFA over DPO with and without considering text diversity on the PartiPrompt dataset. This dataset belongs to a different domain from the training set of Pick-a-Pic v2, and the results are shown below:
>
> | Method          | PickScore Win Rate |
> |------------------|----------|
> | w/o Diversity | 68.2%   |
> | w Diversity    | **71.7%**   |
>
> Detailed results are presented in ${\color{blue}\text{Figure 14 (a)}}$ of ${\color{blue}\text{Appendix L}}$. These results indicate that considering diversity leads to improvements on more prompts. Moreover, the **impact of text diversity would likely be even more significant if the original dataset contained a higher proportion of duplicate or similar prompts**. We further validate this claim with pilot experiments: we create a subset of the Pick-a-Pic v2 dataset by selecting prompts that share similar or same keywords and compare FiFA under different levels of diversity (using different gamma values, higher gamma leads to more diversity). The results are shown below:
>
> | Method            | 100      | 300    | 500    |
> |---------------------------|----------| -------| -------|
> | FiFA (gamma=0)    | 21.238  |  21.175 | 21.056 |
> | FiFA (gamma=0.5)  |  21.377 | 21.510 | **21.563** |
> | FiFA (gamma=1.0)  |  21.387 | 21.513 | 21.560 |
>
> Considering that a higher gamma leads to greater diversity, the results demonstrate the necessity of text diversity when the original dataset contains highly duplicated or similar prompts, as additional training leads to decrease in performance if we do not consider text diversity.

---

> ### Author Response · Authors · 2024-11-25
> **Kind reminder to Reviewer uZ1n**
>
> Dear Reviewer uZ1n,
>
> Thank you again for your time and efforts in reviewing our paper.
>
> As the discussion period draws close, we kindly remind you that two days remain for further comments or questions. We would appreciate the opportunity to address any additional concerns you may have before the discussion phase ends.
>
> Thank you very much!
>
> Many thanks,
> Authors

---

> ### Author Response · Authors · 2024-11-27
> **Gentle Reminder to Reviewer uZ1n**
>
> Dear Reviewer uZ1n,
>
> Thank you once again for your time and effort in reviewing our paper. We greatly appreciate your valuable feedback and suggestions.
>
> We would like to gently remind you that the discussion period is coming to a close.
>
> In our rebuttal, we have:
>
> - **Incorporated feedback on the equation in the revised PDF.**
> - **Clarified points in the introduction.**
> - **Explained that we do not use pretrained models for the preference margin.**
> - **Elaborated on the roles of text quality and diversity.**
>
> If you have any remaining concerns, please do not hesitate to share them with us. We are more than willing to address them promptly.
>
> Thank you very much for your consideration.
>
> Best regards,
> Authors

---

> > ### Author Response · Authors · 2024-11-29
> > **Gentle Reminder to Reviewer uZ1n**
> >
> > Dear Reviewer uZ1n,
> >
> > We truly appreciate your time and effort in reviewing our work.
> >
> > As the discussion period is nearing its end, we kindly remind you that only a few days remain for further comments or questions.
> >
> > In response to your feedback, we have provided a detailed response and added a summary of our response recently.
> >
> > We kindly ask if you have any additional concerns or questions that we may address during the remaining discussion period.
> >
> > Thank you once again for your valuable insights.
> >
> > Best regards, \
> > The Authors

---

> ### Author Response · Authors · 2024-12-02
> **Final Kind Reminder: Review Discussion Period – One Day Remaining**
>
> Dear Reviewer uZ1n,
>
> We wanted to kindly remind you that **only one day remains** in the review discussion period.
>
> We hope that our response and revised manuscript have provided the necessary information to address your questions. If you have had a chance to review our response, we would greatly appreciate it if you could confirm this and let us know if there are any additional questions or concerns that we can address before the discussion period concludes.
>
> If there are no further questions, we hope our revisions and responses have satisfactorily addressed your feedback and would be grateful if this could be reflected in your re-evaluation.
>
> Thank you once again for your time and consideration.
>
> Best regards, \
> The Authors

---

### Author Response · Authors · 2024-11-20
**Summary of Paper Revision**

Thank you all for taking the time and effort to review our paper and provide thoughtful and constructive feedback. We have individually addressed each reviewer’s comments and uploaded a revised version of the paper, which includes some additional results and illustrations:

- ***Writing Improvements***: We added equation numbers, clarified the detailed settings for using both PickScore and HPSv2 in the caption of ${\color{blue}\text{Table 1}}$, and revised the "Analysis of Each Component" section in ${\color{blue}\text{Section 4.3}}$ (page 9).
- ***Limitations Section*** (page 15): We included a discussion on extending our method, such as applying it to DPO variants or other RLHF methods.
- ***Appendix L*** (page 22): We added results in this section with ${\color{blue}\text{Figure 14}}$ to address your concerns, including:
  - Quantitative results on the importance of text diversity,
  - Results when applying our algorithm to a dataset where extreme highest-margin examples are filtered, and
  - Additional results on the ImageReward preference model [1].

Thank you again for your valuable feedback. If you have any further questions or concerns, please feel free to share them. We are happy to engage in further discussions.

---

### References
[1] Xu et al. “ImageReward: Learning and Evaluating Human Preferences for Text-to-Image Generation,” NeurIPS 2023.

---

### Meta-Review · Area_Chair_pqvP · 2024-12-22

**Metareview:**

This paper proposes a method to automatically select high-quality data for diffusion DPO process, which significantly accelerates the training and reduces the GPU hours. With solid experiments, the paper shows its significance in real-world practice. However, the idea of the paper is a common practice in data selection. Therefore, I recommend to accept this paper as a poster, and I also recommend the authors to add more discussions on data selection works.

**Additional Comments On Reviewer Discussion:**

In the initial reviews, the reviewers mainly raise concerns on the following aspects:
1.	unclear writing,
2.	the effectiveness of the reward model,
3.	generalization ability of the proposed method
4.	more ablations on the multiple RLHF objectives.
The authors make clarifications on the writing problems in the rebuttal, conduct more experiments to show how each objective influences the final results, and apply the method to SD3 model to show its generalization ability.
The idea to select data based on whether it contains noise and its diversity is a common practice in the field of data selection, and this paper applies it to the field of DPO process of text-to-image diffusion model.

---

### Decision · Program_Chairs · 2025-01-22

Accept (Poster)